

# Metamorphism of Arctic marine snow during the melt season. Impact on albedo

Gauthier Verin [1,2], Florent Dominé [2], Marcel Babin [2], Ghislain Picard [1], and Laurent Arnaud [1]

[1]UGA, CNRS, Institut des Géosciences de l'Environnement (IGE), UMR 5001, Grenoble, F-38041, France
[2]Takuvik Joint International Laboratory, Laval University (Canada) - CNRS (France), Département de biologie et Québec-Océan, Université Laval, Québec, Québec G1V 0A6, Canada

**Correspondence:** Gauthier Verin (Gauthier.Verin@takuvik.ulaval.ca)

**Abstract.**

The energy budget of Arctic sea ice is strongly affected by the snow cover. Intensive sampling of snow properties was conducted near Qikiqtarjuak in Baffin Bay on typical landfast sea ice during two melt seasons in 2015 and 2016. The sampling included stratigraphy, vertical profiles of snow specific surface area (SSA), density and surface spectral albedo. Both seasons feature four main phases: I) dry snow cover, II) surface melting, III) ripe snowpack and IV) melt pond formation. Each of them was characterized by distinctive physical and optical properties. Highest SSA of $49.3\,\mathrm{m^2kg^{-1}}$ was measured during phase I on surface windslab together with a high broadband albedo of 0.87. The next phase was marked by alternative episodes of surface melting which dramatically decreased the SSA below $3\,\mathrm{m^2kg^{-1}}$ and episodes of snowfall reestablishing the pre-melt conditions. Albedo was highly time variable especially in the near-infrared with minimum values around 0.45 at $1000\,\mathrm{nm}$. At some point, the melt progressed leading to a fully ripe snowpack composed of clustered rounded grains in phase III. Albedo began to decrease in the visible as snow thickness decreased but remained steady at longer wavelengths. Moreover, its spatial variability clearly appeared for the first time following snow depth heterogeneity. The impacts on albedo of both snow SSA and thickness were quantitatively investigated using a radiative transfer model. Comparisons between albedo measurements and simulations show that our data on snow physical properties are relevant for radiative transfer modeling. They also point out to the importance of the properties of the very surface snow layer for albedo computation, especially during phase II when several distinctive layers of snow superimposed following snowfalls, melt or diurnal cycles.

## 1 Introduction

Sea ice features and dynamics in the Arctic are undergoing radical changes, including a shift from multi- to first-year ice (Comiso, 2012), a decrease in thickness (Kwok and Rothrock, 2009) and September aerial extent (Comiso et al., 2008), and an earlier break-up and a later freeze-up (Arrigo and Van Dijken, 2011). These changes strongly affect air-sea interactions (momentum, heat, gases) with multiple feedback loops involved (Serreze and Barry, 2011; Stroeve et al., 2012). They also affect marine ecosystems by substantially increasing the amount of sunlight that penetrates into the ocean and supports photosynthesis under sea ice, and in the open ocean during the ice-free-season now being longer (Arrigo and Van Dijken, 2011).



The snow cover plays a significant role in Arctic sea ice dynamics. Indeed, during winter and early spring, dry snow reflects up to 90% of incoming solar radiation which drastically reduces the energy absorbed by the underlying sea ice (Grenfell and Maykut, 1977; Grenfell and Perovich, 2004; Perovich et al., 1998; Nicolaus et al., 2010). Snow delays the onset of sea ice melt while its albedo remains high enough, and thus directly drives the duration of the melt season which is itself related to

minimum sea ice extent in September (Perovich et al., 2007). The snow optical properties also control the amount of light reaching the upper ocean column under sea ice, much more so than sea ice itself (Grenfell and Maykut, 1977; Perovich, 1990). It was recently showed that major phytoplankton blooms can take place under sea ice (Arrigo et al., 2012), and that snow is the main driver of bloom onset (Mundy et al., 2009, 2014).

The albedo of snow first depends on the optical grain size, the density, the thickness and the impurity content of the snow

(Warren, 1982; Bohren and Barkstrom, 1974; Aoki et al., 2003). Snow is a highly scattering medium composed of ice particles that are weakly absorbing in the visible range (Picard et al., 2016a). Briefly, albedo increases when snow particles size decreases, and these changes are larger in the infrared than in the visible. A snow grain size metric relevant for optical studies is the optical diameter, i.e. the diameter of spheres having the same surface to volume ratio as the snow (Grenfell and Warren, 1999) and at present this is obtained by measuring the snow specific surface area (SSA), both are linked with the relation-

ship SSA$= 6\!\left/\rho_{ice}d_{opt}\right.$. Domine et al. (2006) fairly easily in the field using infrared reflectance methods, for example with 1310 nm radiation (Gallet et al., 2009). Despite the fact that snow SSA is now regularly measured on continental snowpacks, measurements on sea ice are very scarce and limited to a few profiles (Dominé et al., 2002; Domine et al., 2012), limiting our understanding of the albedo of snow on sea ice.

Once deposited on sea ice, snow grains undergo continuous transformations known as snow metamorphism which is mostly

driven by meteorological conditions. While the snowpack is dry, the main factors responsible for metamorphism are the temperature gradient in the snowpack and wind. The temperature gradient between the warmer sea ice and the colder atmosphere leads to an upward water vapor flux coupled to sublimation/condensation cycles that lead to grain growth and the formation of faceted crystals and ultimately large hollow depth hoar crystals (Colbeck, 1983). The upward vapor flux also leads to mass loss so that the density of depth hoar layers often decreases over the season (Domine et al., 2016). Wind, on the contrary, leads

to snow drifting and the sublimation and fragmentation of grains so that wind processes produce hard dense wind slabs made of small and mostly rounded grains. Since the temperature gradient is greatest during the beginning of the snow season, depth hoar formation is more likely then, and Arctic snowpacks usually feature basal depth hoar layers topped by wind slabs and occasionally fresh snow before it gets wind-blown (Dominé et al., 2002; Domine et al., 2012; Sturm et al., 2002; Barber et al., 1995; Langlois et al., 2007; Gallet et al., 2017). Overall snow thickness ranges from a few centimeters up to 70 cm depend-

ing on the roughness of the underlying ice (Sturm et al., 2002) with an average density of 375 $\mathrm{kgm^{-3}}$. When melting starts in spring, snow grains become rounded and daily freeze/thaw cycles leads to rapid grain growth and to the formation of hard dense refrozen layers made of large rounded grains (Colbeck, 1973). In general, snow metamorphism leads to decreases in SSA (Legagneux and Domine, 2005) and consequently in albedo (Picard et al., 2016b; Domine et al., 2006). Snowpack properties therefore vary over time. Given the large wind-induced spatial heterogeneity (Filhol and Sturm, 2015), the snowpack on sea





ice shows large time and space variability which makes the field study of snow properties and in particular albedo challenging, because they require a lot of samplings over a representative time period.

The snow melt period leads to major and sudden changes over sea ice. It extends from the first surface melt event to the formation of melt ponds with typical durations ranging from 10 days to one month (Sturm et al., 2002; Perovich et al., 2002,

2007, 2017). It can be triggered by weather conditions such as positive temperatures or rain events (Sturm et al., 2002; Perovich et al., 2002; Nicolaus et al., 2010). Surface melting results in the formation of a thin surface layer of rounded grains which tends to thicken with further melting. Snow reaches melting temperature as it undergoes wet metamorphism. Once at $0\,^\circ$C, remaining snow layers melt rapidly (Gallet et al., 2017). As snow grain size increases, albedo drops remarkably in the near-infrared almost doubling the energy absorption in the visible range (Perovich et al., 2002; Nicolaus et al., 2010) and acting as

a positive feedback enhancing further melting. The combined effects of surface melting and atmosphere warming enhance the air moisture content which often produces persistent overcast conditions leading to precipitations (Maksimovich and Vihma, 2012; Mortin et al., 2016). Perovich et al. (2002); Gallet et al. (2017); Perovich et al. (2017) observed sudden increases in albedo after such fresh snow precipitations which then suddenly increase the snow albedo and interrupt the melt progression (Perovich et al., 2002; Gallet et al., 2017; Perovich et al., 2017). Perovich et al. (2002) documented an event that lasted 11

days and delayed melt pond formation. Furthermore, sufficient summer snowfalls occasionally allow the snowpack to subsist through the entire summer (Warren and Rigor, 1999). Melt onset appears to be a chaotic transient period in the Arctic ocean, so that some climatic conditions can significantly lengthen the melting period.

For the past 20 years, considerable effort has been made to better understand the radiative properties of snow on sea ice and their evolutions across seasons. Snow albedo drives processes which control the energy budget of sea ice, and albedo

itself depends of snow properties. But studies which aim to link physical and optical properties of snow still remain more or less qualitative. Today, data are lacking to fully quantify and model the global radiative transfer of sea ice because we do not have time series of the snow properties required to understand albedo. Moreover, present data sets do not include systematic combined measurements of snow optical and physical properties at the same spot. This lack of data is particularly detrimental during the melting period when albedo is highly time-variable usually following alternation of freezing and melting events.

The purpose of this paper is to contribute to filling these gaps. We simultaneously documented the temporal evolution of snow physical properties and albedo during two melting periods (2015 and 2016) on typical Arctic landfast sea ice on the east coast of Baffin Island. One or two snowpits were sampled each day or every other day. Measurements included, for the first time over sea ice, the time evolution of the SSA vertical profile, and the corresponding spectral albedo. In addition, the stratigraphy and snow density vertical profile were documented. We first aim at linking surface conditions characterized by

snow physical properties (mostly SSA) with albedo. The second objective is to verify that physical measurements enable reliable simulations of the albedo of snow covered sea ice, especially during the chaotic melting period. To this end, vertical profiles of snow properties are used as inputs to a radiative transfer model. Calculation and albedo measurements are compared and discrepancies are analyzed by assessing the impacts of both snow SSA and snow cover thickness.



## 2 Materials and methods

### 2.1 Study area

Field sampling was conducted close to Qikiqtarjuak Island in Baffin Bay (Figure 1) from May 12 to June 18 in 2015 and from May 17 to June 25 in 2016 as part of the GreenEdge project (Oziel et al., 2019). All measurements were made on typical

landfast sea ice a few hundred meters around an ice camp (same location for both years). In 2015 the ice was very smooth whereas in 2016 the ice surface was disturbed with small reliefs and ridges because of an early break up in December 2015. Surprisingly, as melt ponds formed, vestiges of this early break up could be seen on picture taken from a drone (Figure 2d). The period chosen for sampling corresponds to the snow melting period. Complete snow melt-out and pond formations were only observed in 2016. A meteorological station was set close to the ice camp and provided continuous measurements of 2-m

air temperature and of snow thickness (Oziel et al., 2019).

### 2.2 Albedo measurements

Albedo measurements were performed with a custom-built radiometer (Solalb, developed at IGE following Picard et al. (2016b)). Light was collected using a cosine collector and guided through an optical fiber to a spectrometer (Maya 2000 PRO, Ocean Optics). Irradiance was measured at wavelengths ranging from 300 to 1100 nm, with a 3 nm resolution. More

details about the cosine collector can be found in Picard et al. (2016b). The cosine was fixed at the end of a 3-meter aluminum pole which rested on a tripod 70 cm above the surface. At the other end, the operator manually controlled the arm and triggered the spectrometer. The horizontality was ensured by the operator within less than $0.3\,°$ using an electronic inclinometer mounted next to the cosine collector. Albedo determinations required two sets of measurements for reflected and incident light. Measurements of upwelling and a downwelling irradiance were made sequentially using the same cosine collector with the pole

being manually rotated by $180\,°$. 10 spectra are automatically acquired for each measurements (upwelling and downwelling). No absolute or relative calibration was needed, but measurements had to be made under steady incident light conditions during the 30 s of the process, which seldom strictly prevailed during the Arctic spring. The setup therefore included a reference photodiode to measure light fluctuations at all times for subsequent correction. For both upwelling and downwelling irradiance measurements, the integration times was automatically adjusted in order to optimize the signal to noise ratio. A single operator

could manage the entire process including albedo measurements along linear transects.

### 2.3 Snow physical properties

Here, snow physical properties refer to temperature, snow grain shape and geometric size, SSA and snow density. We first identified the main stratigraphic layers by visual inspection. For each layer, the average snow grain size and shape were determined using a hand lens. Snow temperature was measured at several depths from the bottom of the cover to up to 10 cm

beneath the surface. Freeboard was reported when negative (when sea level was above the interface between snow and sea ice). The vertical profile of snow density was measured using a $100\,\mathrm{cm}^3$, 3 cm high box cutter. The collected snow sample was



weighted using an electronic scale. According to Conger and McClung (2009), this method allows snow density measurements with an uncertainty of 11%. The main uncertainties concern the real volume extracted by the cutter depends on the type of snow. The density of superimposed ice layers was also measured when it was possible. Finally, vertical profiles of SSA were determined from the snow IR reflectance using the DUFISSS instrument (Gallet et al., 2009). Briefly, DUFISSS measures the

albedo of a cylindrical snow sample 63 m in diameter and 25 mm thick at 1310 nm with an integrating sphere. The SSA is deduced from the albedo using a polynomial relationship. The correction concerning the determination of SSA of wet snow introduced by Gallet et al. (2014b) was not applied in this study because it did not induce significant changes on albedo simulations at the end. Uncertainty in SSA determinations is 10% under good conditions (Gallet et al., 2009). Melting can occur if the sample is not handled fast enough, which leads to a lowered SSA value. That is a recurrent issue we had to

deal with after melt onset. Special care was taken to keep every sampling tools as cold as possible, for example by placing instruments in bottom snow layers when the surface was melting.

## 2.4   Sampling Protocol

Data presented in this study were collected either in snowpits or along transects. Snowpits: Albedo was measured first since it requires a pristine area. A minimum of 3 measurements were made depending on sky conditions and light variations. All of

them were performed facing the sun to avoid any shadow from the operator and the equipment. All stratigraphic measurements were carried out along a one meter long shaded trench. Our objective was to conduct all samplings at the same place in order to fully characterize physical and optical properties of the snow at each station. One or two snowpits (requiring three hours of work each) were sampled each sampling day. Fewer snowpits were sampled in 2016 (10 versus 35 in 2015) because the snowpack was already ripe (i.e isothermal at $0\,°C$ and melting throughout) before sampling operations. Snowpit locations were

randomly chosen around the ice camp. Particular efforts were made to sample the widest range of snowpack depth possible in order to catch spatial variability. Transects: Albedo was also measured every 5 m along transects (from $100\,m$ to $150\,m$ long) in order to catch the spatial variability. All the equipment was placed on a sled to make the transport of equipment easier between each measurement station.

## 2.5   Data processing

Upwelling and downwelling irradiance raw acquisitions require several processing steps before the albedo can be obtained. During the field campaigns, spectra were visually checked at the end of the sampling day. Unrealistic data, based on qualitative criteria, were rejected. The first step of processing was to remove the systematic offset in both acquisitions caused by dark current and stray light effects. This offset was approximated for each acquisition as the mean signal at low wavelength (between $200\,nm$ and $260\,nm$), because there is no incoming photon in this wavelength range. Dark current was assumed to be constant

over the entire wavelength range. Then, spectra were divided by their corresponding integration time. Our cosine collectors have been previously characterized on an optical bench in order to assess their exact angular response (Picard et al., 2016b). This angular response was then used to correct the upwelling irradiance measurements depending on the sun zenith angle (SZA) during the acquisition. We excluded any acquisition for which the reference photodiode signal varied by more than





2% between the upwelling and downwelling irradiance measurements. Below 2%, spectra were rescaled using the reference photodiode signal assuming that changes in incident light were equivalent over the entire wavelength range. After all these steps, albedo was calculated as the ratio of upward to downward irradiance. Each upward and downward spectrum is the result of the averaging of a set 10 spectra. Albedo spectra were finally smoothed using a low-pass filter. For each measurement site, it

was checked that all spectra correctly overlapped before being averaged. For the 2015 dataset, the average standard deviation of all integrated albedos (over the 400-1000 nm wavelength range) measured at each snowpit is 0.3% with a maximum of 1%. Thus, in most cases, it is reasonable to assume that the precision on albedo measurements is below 1%.

## 2.6 Radiative transfer modeling

Albedo numerical simulations were performed using the Two-stream Analytical Radiative TransfEr In Snow (TARTES) model
(Libois et al., 2013). Briefly, TARTES uses the delta Edington approximation (Jimenez-Aquinoa and Varela, 2005) in a layered plane parallel snowpack. Each layer is characterized by an average SSA and density. TARTES solves the radiative transfer equation at all depths. For our analysis, only albedo will be presented. Calculations were made using the ice refractive index presented by Picard et al. (2016a). Snow is considered to be impurity free. This assumption has no impact on the results in the near infrared (NIR) where impurity effects is known to be negligible (Warren and Wiscombe, 1980) in comparison with snow
SSA effects (Bohren and Barkstrom, 1974). However, this assumption may impact the albedo at shorter wavelengths (visible range) and this will be assessed and discussed in section 3.4. The underlying sea ice is not modeled, only its albedo (measured on the field) is specified at the bottom of the snowpacks. Albedo depends on solar zenith angle and cloud cover, but a fully diffuse radiation is equivalent to a direct radiation with a SZA of $\sim 50\,°$ (Warren, 1982). In our case, SZAs are between $47\,°$ and $57\,°$, therefore simulations were performed considering a diffuse radiation (SZA of $53\,°$ in TARTES). Doing so, the maximal
error on albedo is $\sim 0.01$ at $1000\,\text{nm}$. The use of TARTES allows the calculation of albedo on a wide wavelength range which makes possible the assessment of broadband albedo and total energy absorbed by the sea ice-ocean system A, in $\text{Wm}^{-2}$. Both were calculated as follows:

$$\alpha = \frac{\int_{300}^{3000} \alpha_s(\lambda) I(\lambda) \mathrm{d}\lambda}{\int_{300}^{3000} I(\lambda) \mathrm{d}\lambda} \tag{1}$$

$$A = \int_{300}^{3000} (1 - \alpha_s(\lambda)) I(\lambda) \mathrm{d}\lambda \tag{2}$$

where $\alpha_s$ is the spectral albedo calculated with TARTES over the 300-3000 nm wavelength range and $I(\lambda)$ is the spectral solar irradiance in $\text{Wm}^{-2}\text{nm}^{-1}$. The solar irradiance spectra was calculated with SBDART, it is representative of solar irradiance observed in Qikiqtarjuaq on June 1st at 12:00 under typical atmospheric conditions of Arctic spring on snow covered areas. The date of June 1st was chosen as the median of albedo measurements dates. The corresponding total wavelength integrated irradiance for this date is $784\,\text{Wm}^{-2}$ and it increased from 740 to $800\,\text{Wm}^{-2}$ along the sample period mainly through
the decrease of the solar zenithal angle (from $47.79\,°$ to $43.66\,°$). Only one solar spectrum was used since the aim of the study





was not to investigate absolute radiation and energy budget, but rather broadland albedo which only depends on the spectra variations of the radiation, not the absolute value.

## 3 Results

### 3.1 General evolution and meteorological conditions

Surface conditions changed drastically during both sampling campaigns as depicted in Figure 2. From the first day of surface melting, it took approximately one month for the snowpack to melt entirely. As previously observed by Perovich et al. (2002) and by Nicolaus et al. (2010), as the melting season progressed sea ice surface became darker and spatial variability increased. The time evolution of albedo at $500\,\mathrm{nm}$ and $1000\,\mathrm{nm}$ are presented in Figure 3 and, similarly to Perovich et al. (2002) and Nicolaus et al. (2010), this evolution clearly shows four main stages confirming visual observations in the field. These phases

are defined below.

Phase I : Cold, dry snow (from the first sampling day on May 13 to 24 in 2015). Sea ice was covered by a dry winter snowpack that had not experienced any melting event. Air temperature increased during this phase but remained below $0\,^{\circ}\mathrm{C}$ (Figure 4). A significant snowfall event associated with strong winds occurred before the first sampling day in 2015 (May 8 and 9), building a fresh snow layer at least $10\,\mathrm{cm}$-thick. Temperature in snow was first colder at the surface, or at least at mid-depth,

($-6.5\,^{\circ}\mathrm{C}$) than at the bottom-most layer where temperatures remained fairly steady between $-5\,^{\circ}\mathrm{C}$ and $-4.5\,^{\circ}\mathrm{C}$ in the day time. The subsequent increase in air temperature reversed the temperature gradient within the snow during this first phase.

Phase II : Surface melting (May 25 to June 11 in 2015; from the first sampling day on May 19 to June 9 in 2016). This phase started with the first surface melting event which coincided with the first positive air temperature in 2015 (Figure 4). Coarse rounded grains and wet grains appeared and albedo decreased in the infrared (Figure 3). Air temperature fluctuated around

$0\,^{\circ}\mathrm{C}$ and several snowfalls were observed both years (Figure 4, snowfalls specified only for 2015). Moreover, the weather was cloudier than during the previous phase and heavy fogs were more common in the early morning. These meteorological conditions persisted in the next phases. Overall, snow temperatures gradually increased until the $0\,^{\circ}\mathrm{C}$ isothermal state was reached.

Phase III : Ripe snowpack (June 12 to the last sampling day on June 16 in 2015; June 10 to June 17 in 2016). At this stage,

the snowpack was at the melting temperature and comprised entirely of rounded polycrystals. This phase is characterized by a decrease in albedo over the visible range for the first time of the season (Figure 3 and 5). Snowpack thickness decreased very quickly until melt-out (4 days in 2015, 7 days in 2016) .

Phase IV : Melt pond formation (June 18 to the last sampling day on June 26, 2016). Snowpacks gave way to a mixture of bare ice and melt ponds. The transition between snow cover and bare ice was progressive, because the ice surface was granular

and looked similar to the large wet grains observed in the ultimate stages of snow melt. As previously observed, sea ice was first rapidly flooded by extended shallow ponds before they drained and got their final shape. During our last sampling day in 2016, June 25, a cooling event associated with snowfall temporally froze the ponds (Figure 2F) and increased albedo (Figure 3).



## 3.2 Snow stratigraphy and physical properties

Only physical properties sampled in 2015 are presented here because they cover the main first three phases, unlike in 2016.

Phase I : Cold, dry snow. The observation of 15 snowpits during this phase revealed a dominant stratigraphy composed of three or four main layers. Snow grain types (and main layers) are presented in Figure 5, vertical profiles of SSA and density

are presented in Figure 6 with average values in Table 1. The bottom-most layer (layer I), in contact with the underlying sea ice, was indurated depth hoar formed from a wind slab (Domine et al., 2016; Sturm et al., 2008), as evidenced by the presence of depth hoar crystal embedded in a matrix of small rounded grains, and confirmed by its high density of $372 \pm 51\,\mathrm{kgm^{-3}}$. Its SSA was $8.9 \pm 4.4\,\mathrm{m^2kg^{-1}}$, typical of depth hoar, whether indurated or not (Domine et al., 2016). Generally the upper part of the snowpack consisted of a layer of indurated faceted grains (layer II) with average SSA of $12.1 \pm 1.8\,\mathrm{m^2kg^{-1}}$ and average

density of $409 \pm 40\,\mathrm{kgm^{-3}}$, topped by a wind slab layer (layer III) made of rounded grains characterized by significantly higher SSA values, $33.4 \pm 2.6\,\mathrm{m^2kg^{-1}}$ and lower density $276 \pm 38\,\mathrm{kgm^{-3}}$. Occasionally a layer of dentritic crystals or fragmented particles could be observed at the surface (layer IVa). The highest SSA values were recorded in this layer, $49.3 \pm 5.9\,\mathrm{m^2kg^{-1}}$ on average (see dark red areas at the surface in Figure 6a). Moreover, sublimation crystals (Gallet et al., 2014a) sometimes formed at the surface of the snowpacks. Figure 6 also shows a significant dichotomy in both profiles with layers

I and II characterized by lower SSA and higher density than layer III. Moreover SSA in layer III gradually decreased over time. Overall, snow depth ranged from $15\,\mathrm{cm}$ to $54\,\mathrm{cm}$. Snow dunes were studied on May 19, 22, 23, 29 and June 4. They corresponded to deeper snowpacks, and were composed of layers I and II only. Furthermore, layer II could be divided into two distinct layers of indurated faceted crystals which showed highest densities values, up to $500\,\mathrm{kgm^{-3}}$, topped by a wind slab. All this information is shown on vertical profiles in Figure 5. Smaller features like sastrugi (Figure 2a) and barchan dunes were

currently observed along the sea ice before melt onset. Freeboard was always positive during phase I.

Phase II : Surface melting. First melting was observed on May 26, one centimeter below the surface and coincided with a low SSA layer at that depth (see Figure 6). Overall, surface melting was characterized by the formation of a layer of rounded polycrystals (layer Va) of low SSA ($10.6 \pm 4.1\,\mathrm{m^2kg^{-1}}$ ). Additionally, as melting conditions persisted this layer got thicker and its SSA kept on decreasing to a minimum of $2.6\,\mathrm{m^2kg^{-1}}$ on June 13 (phase III). The alternation of negative and positive

temperature during night and daytime subjected the surface of the snowpack to a diurnal cycle. During daytime, at the surface, bonds between snow grains melted leading to the observation of wet clustered rounded grain which partially (at least near the surface) froze during the following night forming again dry rounded polycrystals and often a thin layer of melt-freeze crust at the surface. Several snowfalls deposited a new fresh snow layer covering layer Va (Figure 5), which then quickly metamorphised. Fresh snow tended to accumulate in depressions instead of on top of dunes. Melting and subsequent refreezing

increased cohesion between snow grains which totally stopped erosion of snow by wind and therefore its transportation. As the weather became cloudier, a thin layer of surface hoar or needle crystals deposited during the night were regularly observed at the beginning of the day before it rapidly melted. The underlying snow layers I and II, unaffected by surface melting, remained nearly unchanged (with SSAs of $10.6 \pm 4.1\,\mathrm{m^2kg^{-1}}$ and $13.8 \pm 6.9\,\mathrm{m^2kg^{-1}}$ and densities of $370 \pm 26\,\mathrm{kgm^{-3}}$ $418 \pm 51\,\mathrm{kgm^{-3}}$ for layers I and II respectively). SSA of layer III ($24.7 \pm 4.3\,\mathrm{m^2kg^{-1}}$) kept on decreasing during phase II (Figure 6) until it



had completely transformed into wet grains (phase III). Ice layers within the snowpack were first observed on May 29 and became more and more common, to the point that they were present everywhere at the end of phase II and several of them could be found in the same snow column. Two main processes of formation were observed: first, melt-freeze crust formed from the melted surface layers that were buried under new snow and then consequently froze within the snowpack. Secondly,

water percolating from water-saturated layers froze lower in the snowpack to form ice lenses in layer III (first appearance on June 6). As melt became more intense, water reached the interface between layer III and II where it stopped its downward percolation (snow grains are larger in layer II than in layer III) and froze. Liquid water did not go deeper than this interface because capillary forces in layer II were weaker than in layer III due to the discontinuity in snow grain sizes in both layers. After June 4, freeboard was regularly negative, likely due the increasing mass of snow at the surface but also due to bottom ice

melt which may not be excluded caused by influx of warmer ocean water.

Phase III : Ripe snowpack. At this stage, the snowpack was only composed of coarse rounded polycrystals with the lowest SSA values recorded ($4.6 \pm 1.2 \, \mathrm{m^2 kg^{-1}}$ in average), it was isothermal (at the freezing point) and its thickness decreased rapidly afterward. Contrary to 2015, a layer of liquid water, up to $10 \, \mathrm{cm}$ thick, was found nearly everywhere between the ice and the snowpack before snow melt-out in 2016. That layer is likely the result of the imbalance between the rapid input of snow melt

water and the low drain capacity of sea ice rather than the result of a negative freeboard (Perovich et al., 2002).

### 3.3  Spectral Albedo

All albedo spectra from the 2015 and 2016 field campaigns are summarized in Figure 7. They are displayed by phase in order to better illustrate their corresponding specific spectral signatures. Mean albedo values at $500 \, \mathrm{nm}$ and $1000 \, \mathrm{nm}$ are specified in Table 2 for each phase.

Phase I : Cold, dry snow. Highest albedo, $0.97 \pm 0.01$ at $500 \, \mathrm{nm}$, were measured above cold winter snow (Figure 3 and 5). Values slightly decreased along this phase almost only in the infrared from 0.80 to 0.70 at $1000 \, \mathrm{nm}$. Spatial variability was low and the lowest albedo, 0.65 at $1000 \, \mathrm{nm}$, was recorded only above snow dunes, where fresh snow did not accumulate.

Phase II : Surface melting. Following the onset of wet snow metamorphism at the surface, albedo declined mostly in the infrared down to 0.45 at $1000 \, \mathrm{nm}$ while it remained high in the visible ($0.95 \pm 0.024$ at $500 \, \mathrm{nm}$). Figure 3 shows large variations

in albedo ($1000 \, \mathrm{nm}$), in particular sudden increases are observed after snowfall (May 28 and June 3, 2015, for instance). These increases brought back albedo to values observed in phase I. Despite the wider range of albedos presented in Figure 7, spatial variability did not evolve during phase II because changes in snow SSA were homogeneous over the sea ice surface.

Phase III : Ripe snowpack. Darker patches were observed (Figure 2c) as snow thickness declined. These observations were confirmed by spectral albedo measurements. Reflectance decreased in the visible range, from 0.95 to 0.65 at $500 \, \mathrm{nm}$, while

comparatively it remained steadier in the infrared, $0.43 \pm 0.042$ at $1000 \, \mathrm{nm}$ (see Figure 7). In Figure 3, the albedo ranges in the visible are getting wider over time showing that space variability appeared in this phase and amplified as snow thickness decreased. This spatial variability in albedo followed spatial variability in snow thickness. Albedo in the visible was lower above thinner snowpacks, and vice versa.



Phase IV : Melt pond formation. The transition between phase III and IV was ambiguous because melt ponds appeared suddenly while snow patches were still remaining. The albedo decreased over the whole spectral range (Figure 3 and 5) down to 0.74 and 0.25 at $500\,\mathrm{nm}$ and $1000\,\mathrm{nm}$ respectively over bare ice, and 0.32 and 0.06 over melt ponds (see spectra in Figure 7.b). Spatial variability was maximal during this stage. The cooling event that brought less than one centimeter of snow

temporarily enhanced albedo (see Figure 3, June 25). This increase was relatively larger in the near-infrared.

## 3.4 Albedo Modeling

Albedo simulations were performed in order to first assess the relevance of the snow properties dataset for radiative transfer modeling purpose and secondly to quantify the importance of the albedo dependence on the snow surface properties and on snow depth. Albedo was simulated for each snowpit using vertical profiles of SSA and density (as presented in Figure 6) as

inputs in the TARTES model. Only data from the 2015 campaign are presented since this dataset is more comprehensive and cover the three first phases nearly entirely. As shown in the previous section, snow grain size impacts albedo mostly in the near infrared while the impact of snow thickness is mainly observed in the visible. Thus, both measurements and simulation results are compared only at $500\,\mathrm{nm}$ and $1000\,\mathrm{nm}$, they are presented in Figure 8 and summarized in Table 3. Figure 8 also shows (star markers) simulations using a hypothetic infinite snowpack with the same physical properties as the snow surface.

As shown in Figure 8 and in Table 3, two types of errors can be identified. First, the spectral albedo is slightly overestimated at all wavelengths by TARTES along the sampling period. The bias is nearly constant at 500 nm (1.3%) during the whole season with a standard deviation (STD) of 0.09. At 1000 nm, the bias is slightly larger (2.0%) and is much more variable along the season (STD of 7.8). Simulations with SSA reduced by 20% (see Figure 8 and Table 3), larger than the expected uncertainty, is not sufficient to offset the bias which is lowered to 1.0% at 500 nm. Therefore we can reasonably estimate that the bias

comes from our albedo measurements rather than from a systematic error on SSA. Following Wright et al. (2014), shadows from the operator and devices may account for around 50% of this bias, reducing the reflected irradiance by 0.4% to 0.7%. In addition, impurity content within the snow may have also contributed to lower albedo in the visible range. If it is the case, the impurities may have potentially lowered the albedo by 1% at maximum at 500 nm which is negligible compared to the effects caused by snow thickness variations (Figure 3 and Figure 7C). These reasons justify the assumption of an impurity-free

snowpack for modeling purpose in this study. Occasional errors, are however found (more distinctive at $1000\,\mathrm{nm}$) mostly during and after phase II. These errors may be due to erroneous measurements of snow properties due to warmer temperature, much larger than 20% on SSA, and by an inappropriate vertical resolution in sampling (every $1\,\mathrm{cm}$ near the surface for SSA). As mentioned in section 3.2, snow properties at the surface changed rapidly during phase II increasing the number of different layers in the first centimeters of the snowpack. However, snow albedo does not only depend on the surface properties but

on the properties of the whole surface layer. The thickness of this layer (h) is related to the light penetration depth (itself a function of wavelength and snow physical properties). In order to quantify h, TARTES was used to simulate albedo of hypothetic snowpacks composed of a surface layer lied on a semi infinite snowpack. Simulations were made at 1000 nm with SSA comprised between $5\,\mathrm{m^2kg^{-1}}$ and $58\,\mathrm{m^2kg^{-1}}$ and with constant snow densities set to $350\,\mathrm{kgm^{-3}}$ for both layers. The thickness of the surface layer was incremented for each couple of SSA until the albedo was fully explained by the surface



layer only with on absolute error of 0.01. The results of these simulations are presented in Figure 9 where h is given for any couple of SSA. They show how determinant the very surface of snow is for its albedo . For snow conditions close to what was observed in phase II (frost or fresh snow with higher SSA $> 20\,\mathrm{m^2kg^{-1}}$ than underlying layers $< 5\,\mathrm{m^2kg^{-1}}$) then h is given to be at least 8 mm. This means that in this case a centimeter sampling resolution may not be sufficient to correctly model

the albedo at $1000\,\mathrm{nm}$. Both, erroneous measurements of snow properties and inadequate sampling resolution may explain the increase of the deviations during phase II between simulations using only the uppermost layer and those using the entire snowpack (star and gray dot markers respectively in Figure 8b). The agreement were better during phase I, which suggest that snow properties were more homogeneous vertically. Note that these observations are only valid for a wavelength of $1000\,\mathrm{nm}$ and that h decreases inversely related to the wavelength. TARTES was also used in order to illustrate how the albedo varies

with snow thickness above bare ice and slush layers at different wavelengths, results are shown in Figure 10. Albedo spectra of bar ice and melt pond (as shown in Figure 7d) were used as soil albedo in TARTES. The objective is to illustrate the albedo decrease in the visible leading to spatial variability as observed in phase III. SSA and density were respectively $3\,\mathrm{m^2kg^{-1}}$ and $400\,\mathrm{kgm^{-3}}$. For both simulations, albedo decreases first exclusively at shorter wavelength as snow thickness decreases. The decline is more significant if the underlying layer is darker. Reflectance at $1000\,\mathrm{nm}$ remains constant as long as snow is

thicker than $4\,\mathrm{cm}$ above a layer of liquid water. Lower values of SSA and density would increase the snow thickness required for the albedo to begin to decline. Albedo measurements at $500\,\mathrm{nm}$ and $1000\,\mathrm{nm}$ are represented in Figure 10, by dots and squares, respectively. They show that albedo decreases with decreasing depth, mostly in the visible range, as it was theoretically predicted. The variations observed at $1000\,\mathrm{nm}$ are more likely the results of the SSA evolution through time rather than the effect of the thinning of the snowpack. Figure 11 shows the evolution of broadband albedo and total solar energy input in sea ice

calculated using albedo simulations presented in Figure 8. Results in phase III are not reliable because corresponding albedo simulations deviate from measurements as mentioned previously. Broadband albedo reached a maximum of 0.87 during phase I and a minimum of 0.77 at the end phase II inducing an increase of the solar input from $100\,\mathrm{Wm^{-2}}$ to $177\,\mathrm{Wm^{-2}}$ under light conditions defined in section 2. The succession of snowfall and melting episodes caused significant variations in solar radiation transmitted to sea ice. The layers of fresh snow reduced the energy input by approximately $20\,\mathrm{Wm^{-2}}$ on May 30 and from

June 8 to 10 (-16% and -11%). While, on the other hand, the rapid metamorphism of the snow resulted in an increase in solar input of $45\,\mathrm{Wm^{-2}}$ from May 30 to June 1st and from June 4 to 6 (+43% and 35%).

## 4 Discussion

The previous section exposed the evolution of physical properties and albedo of snow. It also showed our capability to retrieve these albedo by radiative transfer modeling using collected snow properties. In particular, comparisons between simulations

and albedo measures show less agreement in phase II than in phase I. In this section we aim to first reconstruct the main steps of formation of the snowpacks observed during samplings. Then secondly, the strong temporal variability of albedo in phase II due to changes in snow properties affecting the very surface of the snowpack is discussed as well as the limitations of our measurement protocol intended for radiative transfer modeling.



## 4.1 Snowpack formation

The snowpacks that were observed during the two melt seasons were the results of several month of formation and evolution since the first snowfall over the new sea ice in fall. Snow metamorphism is driven my meteorological conditions, mostly wind and air temperature in winter. Snow physical properties that were sampled carry the signatures of the past conditions and

therefore can be used to reconstruct the main stages of the formation of the snowpacks. The basal layers I and II were formed first. Their constitutions, indurated depth hoar and faceted crystals formed from wind slabs, show that they have experienced metamorphism under high temperature gradient, typical of Arctic conditions, greater than $50\,°\mathrm{Cm}^{-1}$ and perhaps reaching $200\,°\mathrm{Cm}^{-1}$ (Domine et al., 2016). They are characterized by low SSA, always below $14\,\mathrm{m^2kg^{-1}}$ (Table 1), which clearly contrast with SSA of the upper layer III (see Figure 6) which are above $24\,\mathrm{m^2kg^{-1}}$ in phase II and even greater in phase

I. Using the law of decay of SSA under temperature gradient conditions introduced by Taillandier et al. (2007) we found that between 38 days to 62 days were required to observe SSA below $12\,\mathrm{m^2kg^{-1}}$ under various plausible initial conditions (snow temperature from $-20\,°\mathrm{C}$ to $-5\,°\mathrm{C}$ and initial SSA from $50\,\mathrm{m^2kg^{-1}}$ to $70\,\mathrm{m^2kg^{-1}}$). However, these figures could be reasonably inflated because our snow densities are greater than those used by Taillandier et al. (2007) ($300\,\mathrm{kgm^{-3}}$), knowing that metamorphism is less efficient in denser snowpacks (Flanner and Zender 2007). In addition, data from the meteorological

station shows that no significant snowfalls were recorded between mid March and the May 8-9 events (records presented in Figure 4 begins on April 1st). Therefore we can reasonably advance that layer I and II formed before March 15 and also that layer III and IVa formed after to the snowfall episode of May 8-9. Layers I and II also contrast with their high snow density, around $370\,\mathrm{kgm^{-3}}$ for layer I and $410\,\mathrm{kgm^{-3}}$ for layer II, typical of indurated layers formed from wind slabs (Sturm et al., 2008). In addition, layer II may have gotten denser over time because of vapor condensation coming from layer I (Domine et al.,

2016). Comparatively layer III which is less dense ($276\,\mathrm{kgm^{-3}}$) would have formed under relatively lower winds even though occasional wind speeds above 10 ms-1 were recorded on May 8. One other observation may also confirm this hypothesis, the five dunes that were studied show specific characteristics leading us to believe that they were formed under strong wind events. These dunes were mainly composed of layer I and II characterized by low SSA and high density. In all these five cases, layer II could be clearly subdivided into two layers. One is composed of faceted grains with SSA of $10\,\mathrm{m^2kg^{-1}}$ (layer IIa) and a

density of $475\,\mathrm{kgm^{-3}}$ topped by a windslab of SSA of $17\,\mathrm{m^2kg^{-1}}$ and density of $450\,\mathrm{kgm^{-3}}$ (layer IIb). For the same reasons as previously mentioned, these layers were likely formed before March 15. Moreover, layers IIa and layers IIb were formed during two events that were distinctive because boundaries between them were very well defined between 15 - 27 cm deep. Note that these boundaries were not clearly observed in the thinner layers II of conventional snow pits. At this stage, using Figure 6, we can retrieve snow thickness before layer III formed in May. This reveals that thin snowpacks 10 cm thick and

high dunes between 40 cm and 55 cm thick were present at the same time along the sea ice. This high variability in snow depth and such high densities make us believe that these dunes were in fact whaleback dunes that usually formed under very strong winds (15 ms-1) as described by Filhol and Sturm (2015). From all this information we can assume that the snowpack followed three main steps of formation before the sampling period: 1- A first snowfall accompanied with strong winds formed a highly variable snow cover in term of thickness with the presence of prominent whaleback dunes. 2- Subsequent snow falls, before



March 15, formed other layers on the whaleback (layer IIb) and elsewhere. Gradient metamorphism acted all the time leading to indurated depth hoar from basal dense windslab and to faceted grains in intermediate layers. 3- Layer III formed later in May 8 - 9 few days before sampling operations and therefore was still characterized by large SSA values.

## 4.2 Albedo and surface evolution

The results concerning the overall evolution of albedo confirmed previous observations (Perovich et al., 2002; Grenfell and Perovich, 2004; Nicolaus et al., 2010; Gallet et al., 2017). First high in winter albedo gradually decreased, mainly in the near infrared, after melt onset as snow grains coarsened at the surface until the snowpack completely vanished. Albedo was affected by synoptic weather, such as rain and snowfall events. Snowfalls appeared to be common during melting. They suddenly and temporarily increases the albedo, substantially delaying melt-out. Our observations of two melt seasons near Baffin Island

including numerous spectral albedo measurements and observations of detailed physical properties suggest that a new main relevant phase may be distinguishable: phase III. This phase is characterized by spectral albedo signatures and a typical snow stratigraphy. The snowpack was completely ripe and composed of clustered rounded grains and melt-freeze crusts which strongly enhanced the light penetration depth especially at lower wavelengths. As a result, the albedo decreased for the first time in the visible as a function of both snow thickness and optical properties of the underlying surface (Figure 10) rather than

as a function of snow grain size (phase I and II). This observation was also made by Grenfell and Perovich (2004) and Pirazzini et al. (2006), but they did not mention this strong spectral signature because they used broadband albedo rather than spectral albedo. Albedo spatial variability increased and the sea ice surface became patchy following snow depth variabilities. These fundamental changes in stratigraphy should also have strong effects on other aspects of the energy budget of sea ice. As snow became more transparent to shortwave radiations, more energy was able to be absorbed or transmitted through the underlying

ice. Moreover, since the snowpack is isothermal the energy which is absorbed by the snow might be nearly exclusively used as latent heat, enhancing the melt rate and sublimation. After melt onset, most significant changes in albedo were the consequences of synoptic weather such as snowfalls. For example, on June 4 (Figure 8), only one centimeter of fresh snow was sufficient to increase the albedo by 0.20 at $1000\,\mathrm{nm}$. Although at a lesser extent, albedo evolved all the time because snow conditions at the very surface changed all the time. These changes are due to the cooling of the air during the night which thus provokes

light depositions of needle snow, the formation of surface hoar and the freezing of the uppermost layers. We observed an increase in albedo of 0.12 at $1000\,\mathrm{nm}$ on June 10 on a thin layer of needles. A SSA of $21.3\,\mathrm{m^2kg^{-1}}$ was measured, but is likely underestimated because of hard sampling conditions that day. Gallet et al. (2014a) studied the effects of the formation of surface hoar and sublimation crystals on SSA on the Antarctic plateau. They observed an SSA increased from $33.3\,\mathrm{m^2kg^{-1}}$ to $46.3\,\mathrm{m^2kg^{-1}}$ over 24h leading to an albedo increase of 0.12. The daily melt-freeze cycling also affected the surface but

as surface hoar we were not able to measure the effects on SSA. On June 6,8 and 15 we observed a decrease in albedo that coincides with the melting and the humidification of the very surface between the morning and the afternoon samplings but we can not ensure the exact origin of these changes. We think that large discrepancies between measurements and simulations on June 13 and 15 might be due to this daily cycling that we may not have captured with DUFISSS (reasons discussed below). These daily cycles have been studied by Pirazzini et al. (2006) and Meinander et al. (2008), they showed that diurnal variations



of albedo are often larger than daily mean albedo differences between consecutive days. In the end, we can reasonably advance that daily variations in albedo can exceed 0.10 at $1000\,\mathrm{nm}$ as we observed after needle snow deposition.

## 4.3   Albedo modeling, limitations and suggestions

Occasional discrepancies between modeling and albedo samples pointed out in section 3.4 are probably the sum of measurements errors and inappropriate sampling resolution making them challenging to be fully identified a posteriori. Nevertheless, some hypothesis can be put forward concerning the largest errors on May 26, and June 4, 13 and 15. We found that artificially varying snow densities, even beyond expected uncertainty, had only little effects on albedo compared to the effects of varying the SSA. On May 26, the snowpit was performed during the afternoon, shortly after the snow began to fall. It is possible that the thickness of the uppermost layer of fresh snow that has been sampled for SSA measurements was larger than the one present during the albedo measurement because of the delay separating both samplings (at least 1 hour). On June 4, 13 and 15, albedo simulations are well below measurements probably because SSA was underestimated due to warm conditions (see deviation between simulation and observation in Figure 8). Handling snow samples with tools that have been warmed by positive air temperature is sufficient to induce errors in the measurement. Furthermore, a recurrent underestimation was made when a thin ($< 1\,\mathrm{cm}$) surface layer of fresh snow, surface hoar, or refrozen polycrystals topped a layer of wet coarser grains as currently observed during the morning measurements. As measuring the SSA necessarily requires to slightly press a sample of snow into the $2.5\,\mathrm{cm}$-thick DUFISSS container (using a piston, see Gallet et al. (2009)) the liquid water at the bottom of the sample may have altered the thin surface layer. That was the case on June 4 and 15 where the $1\,\mathrm{cm}$ thick surface layer was very challenging to sample for this reason. Figure 2e shows a cross section of snowpit 1 June 4, that illustrate this issue, one can clearly see the thin fresh snow layer lying on a layer of coarse wet grains. In summary, the results of simulations presented in Figure 8 and Table 3 show that physical properties that were measured explain the albedo variations during phase I of cold winter snowpacks well. The larger discrepancies appearing in phase II and III are isolated and mainly caused by erroneous SSA measurement and inappropriate vertical sampling resolution as discussed above. This highlights how important the very surface is in driving the albedo of the whole snowpack (Figure 9). Snow stratigraphy, like the one depicted in Figure 2e, needs special considerations because very different layers composed the first top centimeters. The main issue comes from the layer of wet grains that alters the SSA of the surface during DUFISSS operations. A solution could have been to sample each layer separately by gathering enough snow for DUFISSS containers while precisely measuring the thickness of each layer and to use a thinner snow box cutter for density sampling. But such a protocol would not have guaranteed better results in albedo simulations if any small spatial variability in snow was present in the field of view of the cosine collector.

## 5   Conclusions

Snow over sea ice has been intensively studied during two melt seasons in Baffin bay. These studies include spectral albedo measurements and vertical profiles of physical properties of snow that are relevant to radiative transfer modeling. The entire transition from cold and dry winter snow covers to ponded sea ice was recorded. Both years, albedo evolved following four





main phases related to the conditions of the snow cover. During these phases, Broadband albedo was first high, up to 0.87, over winter snow (phase I) composed of basal layers of indurated depth hoar and faceted crystals topped by one or several windslabs with sometimes a layer of fresh snow. Albedo gradually decreased in the near infrared as snow grain coarsened up because of surface melting (phase II). At some point, the snowpack was entirely ripe and isothermal, its thickness decreased faster and its

albedo decreased in the visible range for the first time (phase III). This albedo behavior is due to the influence of the underlying darker sea ice as light penetration depth in snow increased. Spatial variability appeared and was directly linked to the snow thickness and optical properties of underlying media. Melt pond formed after snow melt-out during phase IV. Snow physical properties were used as inputs to a radiative transfer model in order to simulate the albedo. The comparisons between albedo measurements and simulations showed that our data was relevant to characterize the snow albedo as long as the uppermost

snow layers remained homogeneous vertically as observed in phase I. After melt onset, the agreement is less good because of the increased number of different superimposed layers at the very surface and because of the greater difficulty to take SSA measurements under warm conditions. Overall, the simulations allowed to precisely link the impacts of both snow properties and snow thickness on the spectral variations of albedo. Before melt onset, SSA decreased with depth because deeper layers were older and thus subjected to snow metamorphism over a longer period. After melt onset, the SSA at the surface rapidly

evolved, decreasing to less than $3 \, \mathrm{m^2 kg^{-1}}$ within less than a day because of melting, or increasing to $60 \, \mathrm{m^2 kg^{-1}}$ because of snow falls, which temporarily enhanced the albedo to pre-melt levels and thus delayed snow melt-out. These changes in surface conditions had a significant impact on the total solar irradiance transmitted to the sea ice system. In particular, the rapid metamorphism of the snow could increase solar input by up to 43%. In addition, a diurnal cycle affecting the snow surface was observed and included formation of surface hoar, light snowfall of needles crystals during the night and freezing

of the very surface. In a lesser extent than snowfalls, these cycles may have temporary enhanced the albedo by up to 0.10 at 1000 nm before the surface metamorphised to wet clustered rounded grains during the day. Finally vertical profiles of SSA can be used to identify the main stages of the snowpack formation. Early-season conditions play an important role in inter-annual variabilities. In 2015, whaleback dunes were common and responsible for strong snow depth variability along sea ice while they did not form in 2016 resulting in a smoother snowpack. Such vertical profiles of physical properties is valuable for albedo

modeling but for radiative transfer studies of the whole snowpack as well, and thus, will be used in further works to precisely quantify the importance of snow cover and its impurity content on sea ice as controlling the light transmittance reaching the ocean upper column.

*Author contributions.* GV, MB and GP designed the study, GV, FD and GP conducted the field samplings. GP and LA designed the in-

struments for optical measurements of snow. Analysis of the entire dataset was performed by GV. All authors contributed to writing of the manuscript, led by GV.



*Competing interests.* The authors declare that they have no conflict of interest.

*Acknowledgements.* The GreenEdge project is funded by the following French and Canadian programs and agencies: ANR (Contract #111112), CNES (project #131425), IPEV (project #1164), CSA, Fondation Total, ArcticNet, LEFE and the French Arctic Initiative (GreenEdge project). This project would not have been possible without the support of the Hamlet of Qikiqtarjuaq and the members of the community as

5  well as the Inuksuit School and its Principal Jacqueline Arsenault. The project is conducted under the scientific coordination of the Canada Excellence Research Chair on Remote sensing of Canada's new Arctic frontier and the CNRS & Université Laval Takuvik Joint International laboratory (UMI3376). The field campaign was successful thanks to the contribution of J. Ferland, G. Bécu, C. Marec, J. Lagunas, F. Bruyant, J. Larivière, E. Rehm, S. Lambert-Girard, C. Aubry, C. Lalande, A. LeBaron, C. Marty, J. Sansoulet, D. Christiansen-Stowe, A. Wells, M. Benoît-Gagné, E. Devred and M.-H. Forget from the Takuvik laboratory, C.J. Mundy and V. Galindo from University of Manitoba as well

10 as F. Pinczon du Sel and E. Brossier from Vagabond. We also thank Michel Gosselin, Québec-Océan, the CCGS Amundsen and the Polar Continental Shelf Program for their in-kind contribution in polar logistic and scientific equipment.





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

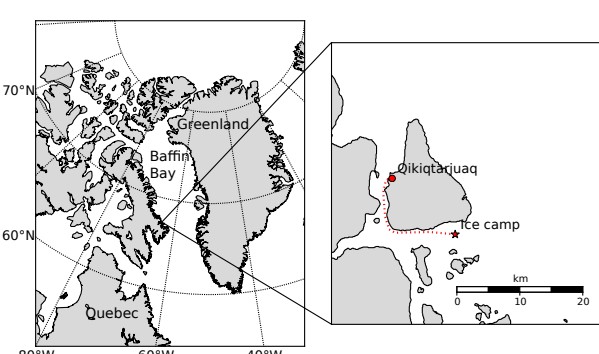

**Figure 1.** Location of the measurement site close to Qikiqtarjuaq island (67 ° 33' 29" N, 64 ° 01' 29" W), east coast of Baffin island, Canada.



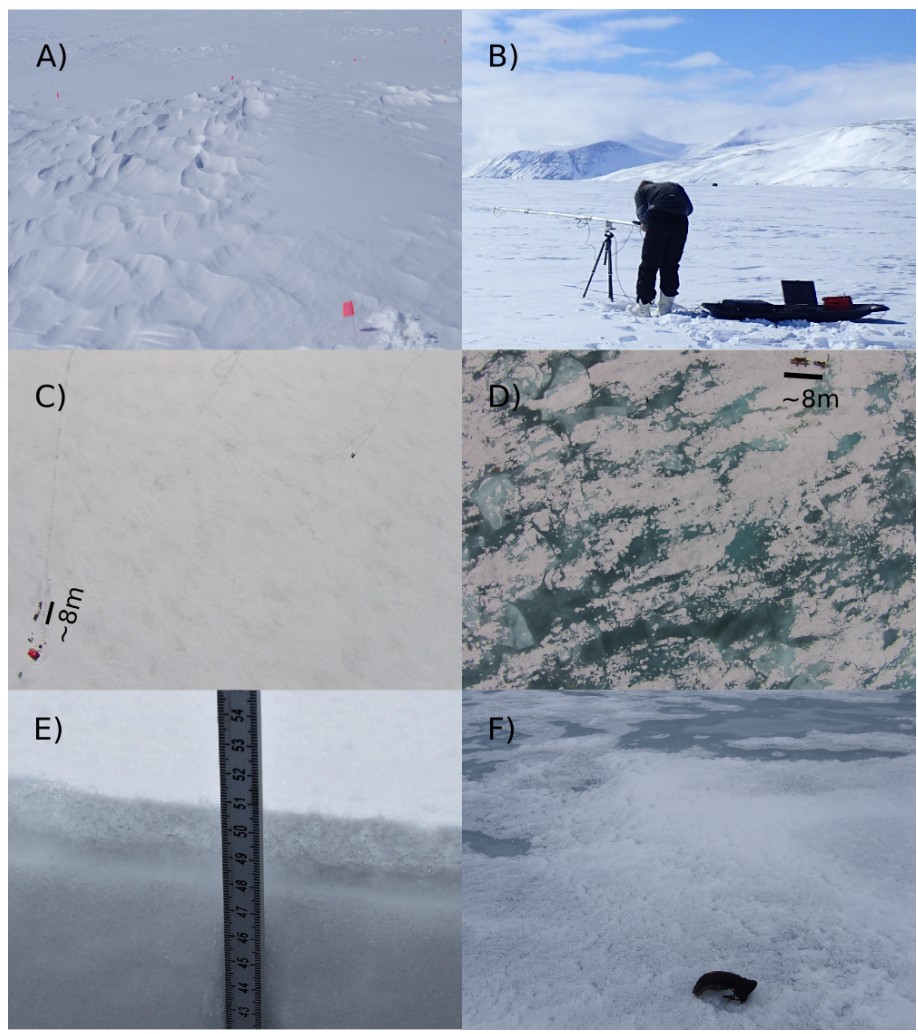

**Figure 2.** Pictures of sea ice at different stages of the 2015 and 2016 melting season : A) dry snow cover, 26 May 2016, B) albedo measurement with Solalb (2015/05/19), C) snow melting, spatial variability begins to be observable (drone picture 2016/06/13), D) melt pond formation (drone picture 2016/06/25), E) Typical snow surface in phase II that was hard to sample, a thin layer (few millimeters of fresh snow covers a 1.5 cm thick layer of wet grains (2016/06/04 snowpit 1) and F) picture of refrozen melt ponds on June 18 (2016) covered by a thin layer of fresh snow ; glove for scale.



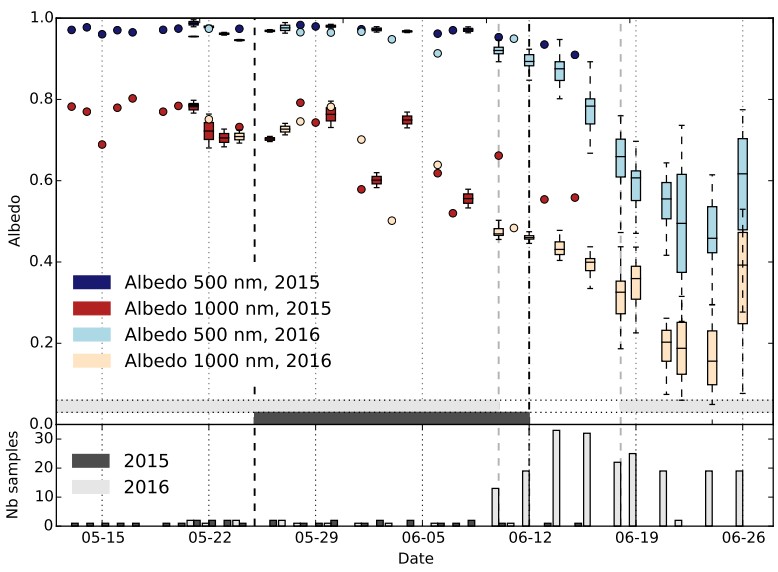

**Figure 3.** Time evolution of albedo at 500 nm in blue and 1000 nm in red for the 2015 and 2016 field campaigns in darker and lighter colors respectively. Boxes are used if more than 1 measurement per day is available, colored dots otherwise. Main phases are specified by horizontal grey bars for both years (Phases I,II and III for 2015 and Phases II,III and IV for 2016). Gray bar graphs at the bottom represent the number of albedo measurements per day.





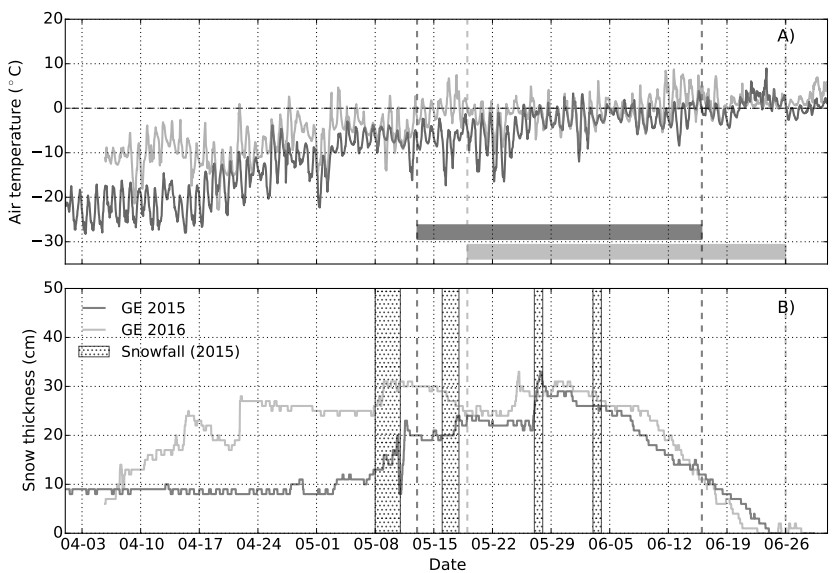

**Figure 4.** Continuous measurements of the sea ice meteorological station. A)Time evolution of air temperature and B) snow thickness measured for GreenEdge 2015 (dark gray) and 2016 (light gray). Gray horizontal bars in (A) denote sampling period for both campaigns. Additionally, main snowfalls in 2015 are specified.

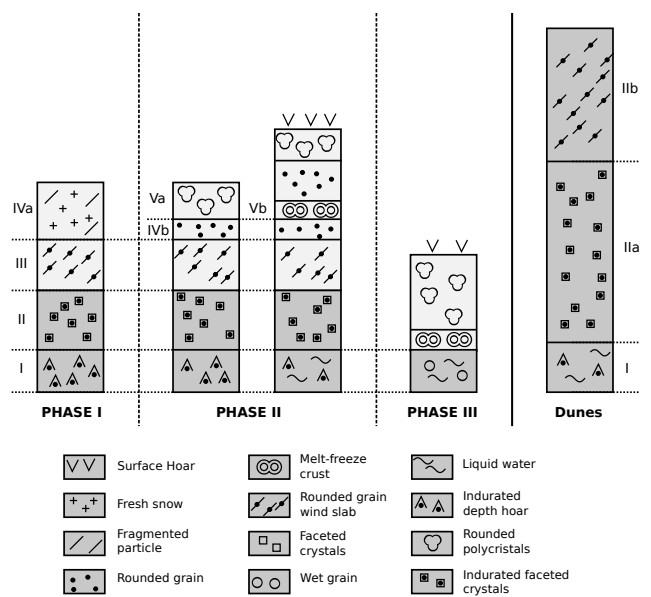

**Figure 5.** Main stratigraphic layers observed for each phase and for snow dunes additionally. Color code is used to distinguish the two main layers according to their SSA values range : dark and light gray for low (layers I and II only) and high SSA values respectively. Vertical scale is not provided as snow depths were highly variable

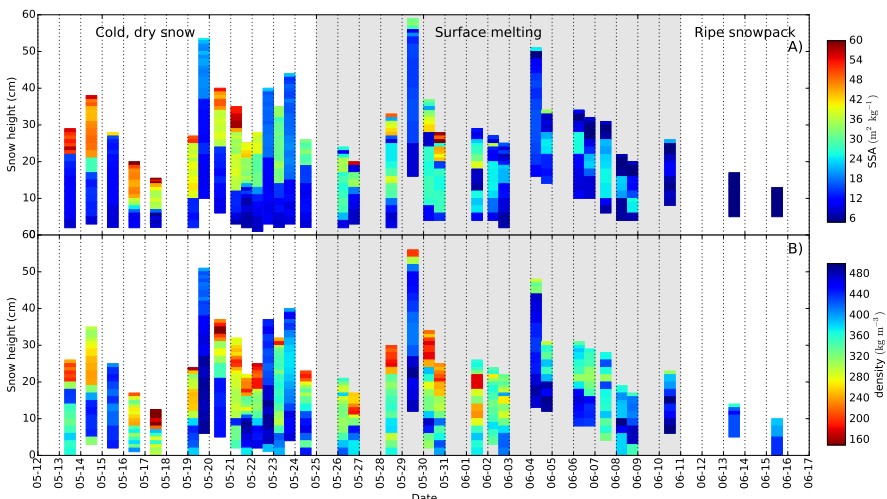

**Figure 6.** Vertical profiles of SSA (A) and density (B) for each snowpit sampled in 2015. Snow elevation (in centimeters) on y axis, sampling dates on x axis. Phases I to III are specified.

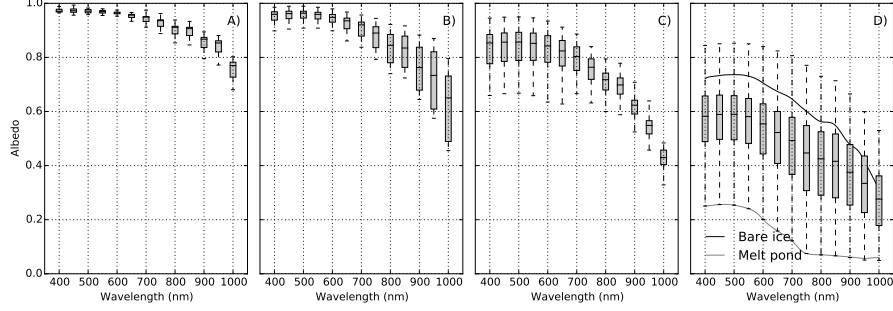

**Figure 7.** Spectral albedo from 400 nm to 1000 nm for both years, represented with boxplot graphs and sorted by phases : A) cold, dry snow, B) surface melting, C) ripe snowpack and D) melt pond formation, here albedo over bare ice only and melt pond only are also shown.

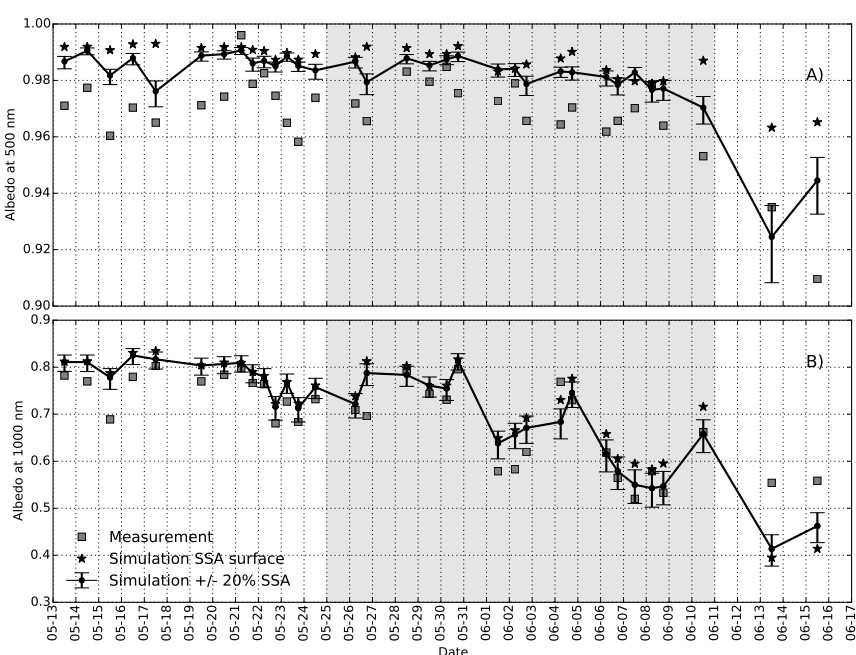

**Figure 8.** Albedo measurements (black) and modeling (gray) at 700 nm (A) and 1000 nm (B) for each sampling station in 2015 (different scale in y axis). Error bars on both sides of simulation points represent results with SSA reduced and enhanced by 20%. Modelings of albedo using the surface layer of the snowpack only (extended as a semi infinite snowpack) are presented with star markers. The grey shaded area specifies the melting period.


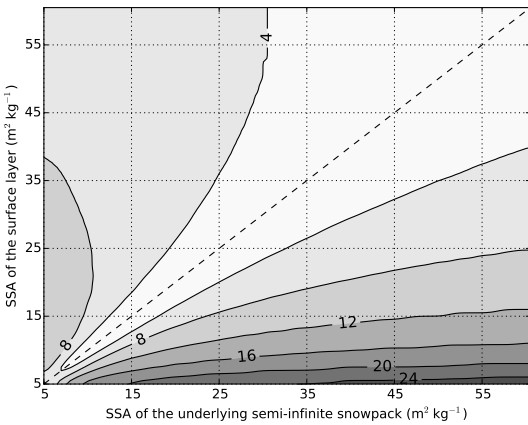

**Figure 9.** Minimal depth (in mm) of a surface snow layer relying on a semi-infinite snowpack, which is required to fully explain the albedo at 1000 nm (error below 0.01). In other words, above this minimal depth the underlying snow layer has no influence on albedo. Results are given for various couples of SSA (from $5\,\mathrm{m^2 kg^{-1}}$ to $58\,\mathrm{m^2 kg^{-1}}$ ) at the surface and within the semi-infinite underlying snowpack. Snow density was set to $350\,\mathrm{kg m^{-3}}$ for both layers.

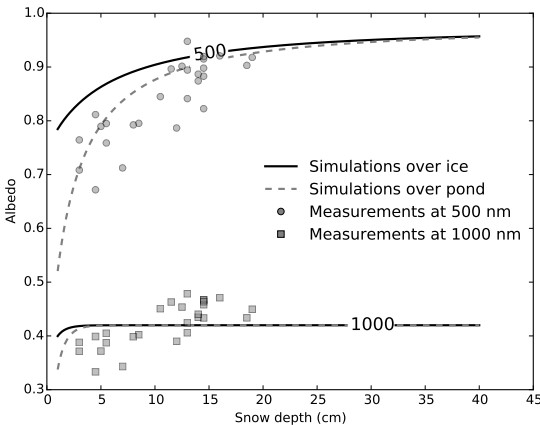

**Figure 10.** Simulations of albedo with varying snow depths. Results are given at 500 nm and 1000 nm. Simulations were performed with a homogeneous snowpack, SSA of $33\,\mathrm{m^2 kg^{-1}}$ covering whether bar ice or a slush layer (solid and dashed lines respectively). Dots and square markers represent the data at respectively 500 nm and 1000 nm collected during phase III in 2016 along two albedo transects (June 13 and 15) where snow depths were also measured.


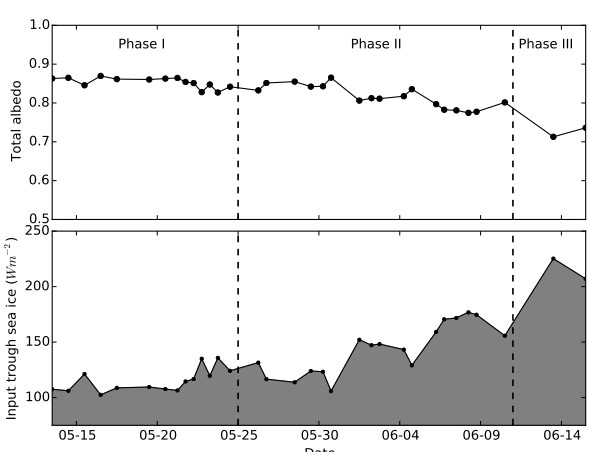

**Figure 11.** Broadband albedo (top) and total energy transmitted to the sea ice system (bottom) in $\mathrm{Wm}^{-2}$, over the 2015 field campaign. Main phases are specified for each graph.





**Table 1.** Average values of SSA and density and corresponding standard deviations for each phase (SSA in $m^2$ $kg^{-1}$ and density in $kg$ $m^{-3}$).

|          | Phase I      |            | Phase II      |           | Phase III |          |
|----------|--------------|------------|---------------|-----------|-----------|----------|
|          | SSA          | Density    | SSA           | Density   | SSA       | Density  |
| Layer I    | 8.9 ± 4.4   | 372 ± 51   | 10.6 ± 4.1    | 370 ± 26  |           |          |
| Layer II   | 12.1 ± 1.8  | 409 ± 40   | 13.8 ± 6.9    | 418 ± 51  |           |          |
| Layer III  | 33.4 ± 2.6  | 276 ± 38   | 24.7 ± 4.3    | 340 ± 49  |           |          |
| Layer IVa  | 49.3 ± 5.9  | 260 ± 122  | 36.3 ± 18.7   |           |           |          |
| Layer IVb  |             |            | 35.0 ± 5.6    | 214 ± 14  |           |          |
| Layer Va   |             |            | 11.6 ± 5.6    | 346 ± 37  | 4.6 ± 1.2 | 406 ± 15 |

**Table 2.** Mean albedo and corresponding standard deviation at 500 nm and 1000 nm along each phase. In phase IV, measurement above ice and pond only (one station for each) are specified.

|                 | Phase I        | Phase II        | Phase III       | Phase IV        |       |       |
|-----------------|----------------|-----------------|-----------------|-----------------|-------|-------|
|                 |                |                 |                 | All             | Ice   | Pond  |
| Albedo 500 nm   | 0.97 ± 0.01    | 0.95 ± 0.024    | 0.84 ± 0.073    | 0.57 ± 0.122    | 0.74  | 0.32  |
| Albedo 1000 nm  | 0.75 ± 0.042   | 0.63 ± 0.121    | 0.43 ± 0.042    | 0.27 ± 0.120    | 0.25  | 0.06  |

**Table 3.** Relative deviations between albedo simulations and measurements in percentages at 500,700 and 1000 nm, and corresponding standard deviations.

|                              | All results  | Phase I     | Phase II    | Phase III     |
|------------------------------|--------------|-------------|-------------|---------------|
| Albedo at 500 nm             | 1.3 ± 0.9    | 1.4 ± 0.8   | 1.2 ± 0.6   | 2.3 ± 2.2     |
| Albedo at 1000 nm            | 2.0 ± 7.8    | 4.4 ± 2.7   | 2.8 ± 6.0   | -21.3 ± 4.0   |
| Albedo at 500 nm, -20% SSA   | 1.0 ± 0.9    | 1.2 ± 0.8   | 0.9 ± 0.6   | 1.3 ± 2.3     |
| Albedo at 1000 nm, -20% SSA  | -2.3 ± 8.3   | 1.4 ± 2.6   | -2.3 ± 6.3  | -27.8 ± 4.2   |