# Peer review of "Metamorphism of Arctic marine snow during the melt season. Impact on albedo"

_The Cryosphere, 2019_

## Referee Comment (RC1) · Anonymous Referee #1 · 18 Aug 2019

General comments: This paper presents results of snow physical properties and spectral albedo measured with high frequency on sea ice in Baffin Bay of Northern Canada during two melt seasons in 2015 and 2016, and result of spectral albedo simulations using radiative transfer model. The authors discuss in detail the evolutions of snow physical and optical properties by dividing the observation period of each year into 4 phases in terms of snow stratigraphy, snow specific surface area (SSA), snow density, and spectral albedos. They also discuss causes of temporal variation of albedos in the visible (500 nm) and near infrared (1000 nm) by comparing the albedos measured with those calculated using radiative transfer model, and by sensitivity tests for effects of SSA and snow layer thickness on albedos.

My overall impression for the manuscript is that the first half part (~Section 3.3) is

detailed but the latter half part is redundant. Although Chapter 3 is "Results", causes of temporal variations of albedos are DISCUSSED with sensitivity tests using radiative transfer model in Section 3.4. In "4 Discussion", similar things as described in "3 Results" are described repeatedly.

Another reason I feel redundant is that each of Sections 3.4, 4.1, 4.2 and 4.3 and Chapter 5 consists of only one paragraph although those paragraphs have a volume more or less one page. These make the findings, discussing points and conclusion of this paper ambiguous. It is needed to explain with a focus on important points and organize the text.

In Section 3.4, it is discussed on uncertainties in measurements of vertical snow physical properties as cause of disagreement in albedo between measurement and calculation in phases I and II. However, it is not discussed on the horizontal heterogeneity. This would be possible cause of the disagreement. Please discuss on this issue.

Bottom figure in Fig.11 are discussed as "total solar energy input in sea ice" in Section 3.4 (p. 11, L19) and "total energy transmitted to the sea ice system" (p. 28, Figure 11 caption). Is looks to be calculated by equation, (1− albedo)*irradiance, where the irradiance = 784 [Wm-2] shown in p.6, L29. If so, this value of the figure is not solar energy input to sea ice, but just solar energy absorbed by media below snowpack. To discuss on the solar energy to sea ice, it should be calculated by (snowpack's transmittance)*irradiance.

Data set of snow physical and optical properties in-situ measured during 2 seasons in the Arctic is very valuable although the manuscript contains ambiguous points and is still disorganized. To be suitable paper for The Cryosphere, the issues mentioned above should be solved.

Specific comments: p.5, L1: "weighted" is a typo for "weighed"

p.5, L6-8 "The correction concerning the determination of SSA of wet snow introduced

by Gallet et al. (2014b) was not applied in this study because it did not induce significant changes on albedo simulations at the end.": Please indicate amount of error in SSA by not applying this correction.

p.5,L26-27 "Unrealistic data, based on qualitative criteria, were rejected.": Please indicate example(s) of the qualitative criteria.

p.5 L32: "upwelling" is a typo for "downwelling"

p.5 L32-33 "sun zenith angle (SZA)": The term "solar zenith angle" is used in many other parts. It is better to unify the technical term.

p.6 L4 "Albedo spectra were finally smoothed using a low-pass filter.": Does this mean the raw spectral albedo varies not smoothly unless a low-pass filter? Please explain the reason.

p.6 L26: This term "SBDART" is first appearance. Please show what does this abbreviation stand for and the reference.

p.7, L2: Please replace "radiation" with "downward radiation".

p.7, L25: Please replace "the snowpack was" with "all snow layers were".

p.9, L29: I think it is better to replace "Reflectance" with "Albedo".

p.10, L4: "Figure 7.b)." is a typo for "Figure 7b)."

p.10, L17 "with a standard deviation (STD) of 0.09.": The value "0.9" is shown in Table 3.

p.10, L22-25 "If it is the case, . . .": I don't understand this discussion on snow impurities. It is described "the impurities may have potentially lowered the albedo by 1% at maximum . . ." which is larger than the values of "reducing the reflected irradiance by 0.4% to 0.7%". If effects caused by snow thickness variations is larger than the impurities effect (1%), the latter (impurity effect) is not negligible.

p.10, L25: It would be better to start a new paragraph from the sentence "Occasional errors, are however found...". Please confirm the cause of disagreement in albedo at 500 nm at the end of the previous paragraph.

p.11, L7 "(star and gray dot markers respectively in Figure 8b)": I suppose the gray dots are measurements. If so, this sentence is "(star and black dot markers ...".

p.11. L9: It would be better to start a new paragraph from the sentence "TARTES was also used in order to illustrate how...".

p.11. L11: "bar ice" is a typo for "bare ice".

p.11. L11 "as soil albedo in TARTES.": General readers don't know underlying surface of snow is assured to be soil in TARTES. Please revise to more general description "as underlying surface albedo in TARTES."

p.11, L14: It would be better to replace "Reflectance" with "Albedo".

p.11, L18 "The variations observed at 1000 nm ...": This discussion is qualitative. More quantitative discussion is needed.

p.11. L19: It would be better to start a new paragraph from the sentence "Figure 11 shows the evolution of broadband albedo...".

p.11. L20 "calculated using albedo simulations": Please show the value of downward irradiance here again.

p.11. L22-23: Please replace "light conditions" with "light illumination conditions".

p.11. L23: Please replace "section 2" with "section 2.6".

p.11, L30: "albedo measures" is a typo for "albedos measured".

4.1 Snowpack formation: This section describes observation results of snow physical properties qualitatively, some of which were shown in Sections 3.1 and 3.2. Dune formations continues from explanations for phases I – III in the first part, that is confusing.

At least discussion on dunes should be made in separate paragraph.

4.2 Albedo and surface evolution: This section also describes temporal variations of observed albedos and their potential causes, that is not enough as conclusion. And, some of which were shown in Sections 3.3. Do you really need redundant sentence as, for example, "Our observations of two melt seasons near Baffin Island including numerous spectral albedo measurements and observations of detailed physical properties suggest that a new main relevant phase may be distinguishable: phase III." (p.12, L9-11).

p.13, L12-13 "which strongly enhanced the light penetration depth especially at lower wavelengths.": This sentence seems to be observation result of this study. However, there is no measurement data of light penetration depth in the manuscript."

4.3 Albedo modeling, limitations and suggestions: This section is almost summary of Section 3.4.

Figure 3: Please indicate the terms "I, II III and IV" in the figure for main phases.

Figure 4, caption L3: "Additionally, main snowfalls in 2015 are specified." Please add "in (B)" at the end of this sentence.

Figure 5, caption: Please indicate the year.

Figure 7, caption: Please indicate the year.

Figure 8, caption: - "Albedo measurements (black) and modeling (gray) at 700 nm (A) and 1000 nm (B)": I suppose "black" and "gray" are opposite. - "700 nm" is a typo for "500nm". - "The grey shaded area specifies the melting period.": Please replace "melting period" with "surface melting" or "phase-II", which is a defined name for this period in this study.

Figure 9: Isn't the value along the 1 by 1 line zero? The value near an origin is not zero.
Figure 10: Dashed line is represented as"Simulations over pond" in the figure, but "a slush layers" in the caption. Please unify the term.

Figure 10, caption: - "SSA of 33 m2kg−1": The value shown in the text is 3 m2kg−1 (p.11, L12). Which is correct? - "bar ice" is a typo for "bare ice". - Please replace "the data at respectively" with "the albedo measurements at respectively"

Table 3, caption: - "at 500,700 and 1000 nm" is a typo for "at 500 and 1000 nm". - "1.3 ± 0.9": The value "±0.09" is shown in the text. (p.10, L17).

---

## Referee Comment (RC2) · Anonymous Referee #2 · 28 Aug 2019

Review of "Metamorphism of Arctic marine snow during the melt season. Impact on albedo" by Verin et al.

**General comment**

In this paper, Verin et al. described and analysed the snow and spectral albedo observations carried out over landfast sea ice in the Baffin Bay during two melt seasons in 2015 and 2016. The main strength of the presented observational dataset is the collocated measurements of snow microstructure and snow spectral albedo, which make this a very rare and unique Arctic dataset. These measurements enabled the use of a radiative transfer model to test if the collected snow observations were suitable to reproduce the observed albedo. Compared to similar studies over continental snow, the conditions were complicated by the spatial heterogeneity of the snow layer, by the presence of melt ponds at the end of the snowmelt period, and by the wetness of the snow in the last phase of the melting. The results revealed the shortcomings of the applied snow measurement technique in those particular conditions, and suggestions to improve the measurement protocol were given. The applied radiative transfer method appeared robust.

Main weaknesses of the paper are:

- Introduction should be compacted, leaving out all those details and concepts that deviate from the main scope of the paper.
- The snow layer identification seems sometimes arbitrary. Due to the large spatial heterogeneity of the snowpack, it is hard to obtain the stratigraphic evolution of the snowpits, except in a broad line. The number of speculations on the evolution of the stratigraphy should be reduced to a minimum.
- It is not always clear to which campaign (year) the results and discussion refer to. This should be clarified, and possibly the differences in snowpack characteristics between the two years should be discussed more extensively (personally I find the large observed differences very interesting, especially from modelling perspective).
- Several linguistic errors were found. A thorough linguistic revision is therefore recommended.

I recommend the editor to publish the paper in The Cryosphere after a major revision. Here below are my detailed comments.

**Detailed comments**

p.2, line 11-12: "Briefly, albedo increases when snow particles size decreases, and these changes are larger in the infrared than in the visible." Please introduce here a brief explanation for this.

p.2, lines 15-16: "Domine et al. (2006) fairly easily in the field using infrared reflectance methods, for example with15 1310nm radiation (Gallet et al., 2009)." Maybe you meant: "Domine et al. (2006) measured SSA fairly easily in the field using infrared reflectance methods, for example with 1310nm radiation (Gallet et al., 2009)."

p.2, line 16: "…snow SSA is now regularly measured on continental snowpacks". Certainly this is not the case. Only very few groups in the world measure SSA, and even fewer (so few that the authors could list them here, as they can be counted with the fingers on one hand) measure it regularly, i.e. not only in dedicated field campaigns. As the authors demonstrate also with this paper, SSA measurements are extremely challenging and expensive (as all manual measurements are), and techniques are evolving all the time.

From p.2 line 19 to p.3 line 17: what is the point here? Please move part of the text into the discussion section, to compare the results of this paper with previous ones, and leave in the introduction only the key message (few lines only), avoiding all the details and concepts that deviate from the focus of the paper. The introduction should explain the relevance of the treated issue and the problems that previous works have left unsolved and this paper will help to solve. Is the message here that the snow processed studied in this work have been studied already a lot? What is then the added value of this work?

p.2, lines 29-30: "Overall snow thickness ranges from a few centimeters up to 70cm depending on the roughness of the underlying ice (Sturm et al., 2002) with an average density of 375kgm−3". Where this value of average density comes from? Please provide a reference. Also, I would say that snow thickness primarily depends on the thickness of the underlying sea ice and on the amount of solid precipitation, and only secondarily on the sea ice roughness.

p.3, lines 7-8: "Snow reaches melting temperature as it undergoes wet metamorphism. Once at 0∘C, remaining snow layers melt rapidly (Gallet et al., 2017)." Please reformulate or (better) remove.

p. 3, line 8: "snow grain size". The authors need to be consistent with the chosen terminology. In previous paragraphs, the evolution of snow microstructure is described in term of SSA. Please stick to it, or then change it to optical diameter in the whole introduction

p.3, line 18: "For the past 20 years, considerable effort has been made to better understand the radiative properties of snow on sea ice and their evolutions across seasons." Effort was made also before (see e.g. SHEBA experiment, Russian ice drift stations, etc…).

p.4, line 27: What do the author mean for "geometric size"?

p.5, lines 2-3: "The main uncertainties concern the real volume extracted by the cutter depends on the type of snow". Correct as "The main uncertainties concern the real volume extracted by the cutter, which depends on the type of snow".

p.5, line 5: Should 63 m be 63 mm

p.5, line 20-22: Please correct as shown: "Particular efforts were made to sample the widest *possible* range of snowpack depth*s*  in order to catch spatial variability.  *For the same purpose,* albedo was also measured every 5 m along transects (from 100m to 150m long) ".

p.6, line 7: "Thus, in most cases, it is reasonable to assume that the precision on albedo measurements is below 1%." Please replace "precision" with "uncertainty". I disagree with this estimation, as it only account for repeatability of the measurements, while many other error sources are in place (e.g. shadow/obstruction of the measurement setup, deviation from perfect cosine response, reflections/shadows from nearby dunes).

p. 7, line20: "several snowfalls". During the observing period in 2015, only two snowfall events were marked in Fig 4. In addition to these, where there other light snowfall events? If so, please mark all of them (see also my comments later)

In relation to Figure 4, I recommend to:

- Zoom the temperature plot between -30 oC and +10 oC
- Specify (in the figure caption) the time interval of the temperature and snow depth observations

- Describe (in the main text) the accuracy of temperature and snow depth measurements. Temperature measurements may be biased of the sensors are not ventilated and heated, and snow depth representativeness should be discussed.

p.7, line 26: please do not refer to Fig 5, as it does not show any albedo data.

p.7, line 26-27: "Snowpack thickness decreased very quickly until melt-out (4 days in 2015, 7 days in 2016)". This mentioned periods correspond to the lengths of phase III in the two years, but according to Fig 4 snow thickness reached 0 only some days after end of phase III. Please refer here to Fig 4.

p.8, line 13: Could the authors clarify what "sublimation crystal" are, without making necessary for the reader to read Gallet et al. 2014?

p.8, lines 14-15: "Figure 6 also shows a significant dichotomy in both profiles with layers I and II characterized by lower SSA and higher density than layer III." Is it so that the snowpits that do not present such a dichotomy (4 of them) are from dunes? If this is the case, please explain it.

In relation to Fig 6, I recommend to mark the snowpits that correspond to the dunes.

p.8, line 18-19: "Furthermore, layer II could be divided into two distinct layers of indurated faceted crystals which showed highest densities values, up to 500kgm−3, topped by a wind slab." Why the uppermost dune layer is called IIb instead of III? Isn't it generated by the same process at the same time?

Generally, it is very difficult to see correspondence between the layers described in fig 5 and the SSA and density profiles shown in Fig 6. The distinction between layer I and II is quite obscure in Fig 6 (do indurated depth hoar (layer I) and indurated faceted crystals (layer II) have same SSA and density?). Also the distinction between layer II and IV is hard to see in Fig 6. It seems to me that the schematic picture of the stratigraphy in fig 5 can be applies to few selected cases, but then most of the profiles are much more complex, also in view of the spatial heterogeneity. I therefore recommend referring to Fig 5 only as an example of the observed stratigraphy, valid for a subgroup of snowpits, and mark in fig 6 what are the snowpits with stratigraphy that follows fig 5. Alternatively, I recommend to mark the 4 layers (e.g. with horizontal black lines) in each snowpit shown in fig 6. Actually, I recommend marking the layers also in case that stratigraphy is shown only for a selection of snowpits.

p.8, lines 28-29: "Several snowfalls deposited a new fresh snow layer covering layer Va (Figure 5), which then quickly metamorphised". Fresh snow layers are not visible from Fig.5. From fig 4 I can see 2 snowfall episodes during phase II in 2015 (28.5 and 4.6), which are associated to a rise in snow thickness, but from Fig 6 I can probably identify 3 cases of high SSA at the surface (compatible with fresh snow): 26.5, 28.5, and 30.5 (second snowpit). On 30.5 (first snowpit), 31.5 and on 4.6 the surface SSA is medium, but much larger than in the underlying layers, hinting to a fast metamorphism of the fresh snow. Was there light snowfall on 26.5 and 30.5? If not, what caused the rise in SSA? Could you please unambiguously identify all cases when there was light or heavy snowfall, to separate them from the cases of surface freezing and needle formation/deposition?

In conclusion, yes, I can see what the authors states, but it should be explain better, referring to fig 4 and 6, perhaps mentioning that light snowfalls on 26.5 and on 30.05 were not marked in fig 4 (if this is the case!)

p.9, lines 1-3:" Ice layers within the snowpack were first observed on May 29 and became more and more common, to the point that they were present everywhere at the end of phase II and several of them could be found in the same snow column." Could the authors show some example, pointing to specific profiles in Fig.6? E.g. on 7.6?

p.9, lines 7-8: "Liquid water did not go deeper than this interface …." Did the authors measure the profile of liquid water content, or is this conclusion based on other considerations or assumptions? Please clarify.

p.9, line 9-10: replace "was regularly" with "became"? Replace "increasing" with "increased"? Replace "but also due to bottom ice melt which may not be excluded caused by influx of warmer ocean water" with "but also possibly due to bottom ice melt which occurs in case of influx of warmer ocean water"

p.9, line 13: remove "afterward".

Caption of Fig. 7: "… and D) melt pond formation, here albedo over bare ice only and melt pond only are also shown." Please break the sentence: "and D) melt pond formation. In D) albedo over bare ice only and melt pond only are also shown."

p.9, line 26: "… brought back albedo" should be "… brought albedo back"

p.9, line 26: "Despite the wider range of albedos presented in Figure 7" Wider than what? Certainly albedo range is not wider than in Fig 3, as data are the same. Maybe the authors refer to a wider albedo range in 7B compared to the range seen in 7A?

p.9, line 27: "spatial variability did not evolve during phase II" Do the author mean that spatial variability did not increase during phase II? Please correct.

p.9, line 30: Replace "(see Figure 7)" with "(see Figure 7C)"

p.10, line 2-4: Replace "(Figure 3 and 5)" with "(Figure 3 and 7D)" and replace "(see spectra in Figure 7b)" with (Table 2)".

p.10, line 4: should "cooling event" be replaced by "snowfall"?

Can the authors provide more descriptions of the melt ponds? How deep they were, where they open or frozen, how large they were, and were they covering the totality (or more than 90%) of field of view of the downward looking head of the spectro-radiometer (having, thus, a radius of more than 3m)? How many melt ponds have been measured? Had they varying characteristics? Are the provided values for bare ice and melt pond albedo some averages? What was then the standard deviation? Also, in the discussion section it would be good to compare with the bare ice and pond albedo measurements carried out in previous studies.

p.10, lines 6-8: "Albedo simulations were performed in order to first assess the relevance of the snow properties dataset for radiative transfer modeling purpose and secondly to quantify the importance of the albedo dependence on the snow surface properties and on snow depth." This concept is badly expressed. Please consider rewriting "Albedo simulations were performed to assess the adequacy of the collected snow observations for radiative transfer modelling, and to quantify the sensitivity of surface albedo to snow surface properties and snow depth."

p.10, first paragraph: please specify which grain shape was applied in the TARTES simulations.

p.10, lines17-18: are STD percentages? I suppose so… then please add the %.

p.10, lines 18-19: "Simulations with SSA reduced by 20% (see Figure 8 and Table 3), larger than the expected uncertainty, is not sufficient to offset the bias which is lowered to 1.0% at 500 nm." Based on the personal experience of some colleagues, who found strong overestimation of SSA using a similar measurement principle (IceCube), I think that in melting conditions the error in SSA measurements done with the applied sampling technique is closer to 100% than to 20%. The fact that the error increases with increasing wetness of the snow, and that it is most severe for the 1000nm than for the 500nm albedo, suggests that there is a problem in the SSA estimation. I recommend replacing the albedo modelled using +-20% SSA with the albedo modelled using +-100% SSA.

p.10, line 25: "Occasional errors": I guess you mean "occasional cases of larger errors"

p.10, lines 26-27: "These errors may be due to erroneous measurements of snow properties due to warmer temperature, much larger than 20% on SSA,…". Please rearrange the sentence as "Errors in SSA much larger than 20% may be due to erroneous measurements of snow properties caused by the melting conditions".

p.11, lines 2-4:"For snow conditions close to what was observed in phase II (frost or fresh snow with higher SSA > 20 m2kg−1 than underlying layers < 5 m2kg−1) then h is given to be at least 8 mm." Please improve the readability of this sentence, e.g. as "For snow conditions close to what was observed in phase II (frost or fresh snow at the surface with SSA > 20 m2kg−1 and underlying layers with SSA < 5 m2kg−1) h is at least 8 mm."

p.11, line4: Please remove "This means that", and put a comma after "in this case".

p.11, lines 7-8: "The agreement were better during phase I, which suggest that snow properties were more homogeneous vertically." In my view, the main reason for the better agreement between modelled and observed albedo is that SSA measurements are easier and, hence, more reliable in case of dry snow.

p.11, line10-11: "Albedo spectra of bar ice and melt pond (as shown in Figure 7d) were used as soil albedo in TARTES." There cannot be snow above an open melt pond. Did the measured albedo spectra correspond to frozen melt ponds or to open melt ponds? They have quite different albedo. Did the authors measured the albedo of slush? Is it equal to the applied albedo of melt pond? These model results may be used to interpret the evolution of albedo during phase III -IV in 2016, but not in 2015 (which is the year illustrated in Fig 6 and 8), where neither slush nor melt ponds were observed.

Caption of Figure 10: "Dots and square markers represent the data at respectively 500 nm and 1000 nm collected during phase III in 2016 along two albedo transects (June 13 and 15) where snow depths were also measured" I think that this 2016 measurements should be mentioned also in the text, when introducing Figure 10, adding some more info. Was the ice thickness measured? Figure 10 suggests that there were quite large spatial differences in the measured albedo spectra of bare ice and melt ponds/slush, isn't it? I think it is worth discussing it.

p.11, lines 11-12: "The objective is to illustrate the albedo decrease in the visible leading to spatial variability as observed in phase III." This is a very badly formulated sentence. Do the authors mean that "The objective is to illustrate how the increase in spatial variability of snow properties causes a decrease in the visible albedo as observed in phase III"?

p.11, line29: please remove the first word ("these").

p.12, line 17: please remove "to" after "layer III and Iva formed after"

p.12, from line 16 to 28: in my opinion this section includes too many speculations and it is very hard to read. After all, the large spatial heterogeneity prevents the possibility to generalize the stratigraphy of single snowpits. I recommend a strong reduction of the text here, leaving only the essential content related to dune formation and spatial heterogeneity. It would be nice to spend more words on the comparison between 2015 and 2016.

p.13, from line 22 onward: the mentioned dates (June 4 and later June 10, 6, 8, 15) all refer to 2015? If so, please specify (referring to Fig 8 for the first mentioned date is not clear enough). If the whole discussion in 4.2 refers to the measurements collected in 2015, when there was no slush, it has to be made clear. Again, were albedo responses to changes in snow properties different in 2015 and 2016 during phase II and III?

In relation to Fig 8, please correct the figure caption as "Albedo measurements (gray squares) and simulations  (black dottes) at 500 nm (A) and 1000 nm (B) for each sampling station in 2015 (different scale in y axis). Error bars on both sides of simulation points represent results with SSA reduced and enhanced by 20 %. Simulations of albedo using the surface layer  (extended as a semi infinite snowpack are presented with star markers. The grey shaded area specifies the melting period."

p.13, line30: "On June 6,8 and 15 we observed a decrease in albedo that coincides with the melting and the humidification of the very surface between the morning and the afternoon samplings but we can not ensure the exact origin of these changes". Please refer to Figs 3 and 8. I can see the morning and afternoon measurements in Figs 3 and 8 on 6.6.2015 and 8.6.2015, but not on 15.6.2015. If you had two measurements on that day, please show it in Figs 3 and 8.

p.13, lines 29-30: "The daily melt-freeze cycling also affected the surface but as surface hoar we were not able to measure the effects on SSA." Grammatically incorrect sentence. Maybe it should be "The daily melt-freeze cycling also affected the surface, but as this cycle often caused the formation of surface hoar, which cannot be sampled for SSA measurements made using DUFISSS, we were not able to estimate the melt-freeze cycling effects on SSA."

p.14, line 6: "…the largest errors on May 26, and June 4, 13 and 15". Please mention the year 2015 (it is obvious for you, but it reduce effort and wondering to the reader). What about the large error on May 15, 2015?

p.14, lines 8-10: "On May 26, the snowpit was performed during the afternoon, shortly after the snow began to fall. It is possible that the thickness of the uppermost layer of fresh snow that has been sampled for SSA measurements was larger than the one present during the albedo measurement because of the delay separating both samplings (at least 1 hour)." I would bet that the main reason for the good agreement with the measured albedo in the morning and the large error in the afternoon is that during the morning the snow temperature was below freezing (at least according to Fig. 4) while in the afternoon snow was melting. Hence, in the afternoon the snow sampling with the cylinder sampler was very difficult, and the resulting SSA measurements were largely overestimated (the reason for this is not clear to me: I think it has to do with the mechanical change of the microstructure when collecting the snow sample).

p.14, lines 13-15: "Furthermore, a recurrent underestimation was made when a thin (< 1 cm) surface layer of fresh snow, surface hoar, or refrozen polycrystals topped a layer of wet coarser grains as currently observed during the morning measurements." I agree. I think that, in snow melting

conditions, the SSA retrieved with the snow sampling method (DUFISSS, IceCube) has often a positive bias in case of a uniformly wet layer, and a negative bias in case of a fine surface layer overlying a wet, coarser grain layer (because of the reason described in the text).

p.14, lines 27-28: "But such a protocol would not have guaranteed better results in albedo simulations if any small spatial variability in snow was present in the field of view of the cosine collector." Please replace "But" with "However," or similar. It is true that better measurement protocol does not guarantee better albedo simulations if point representativeness is low, but I think that the snowpit representativeness and the accuracy of the snowpit SSA measurements should be treated separately, both in the best possible way.

p.15, lines 5-6: "This albedo behaviour is due to the influence of the underlying darker sea ice as light penetration depth in snow increased." I would add "and snow depth decreased".

One final question: is it so that melt ponds did not form in 2015? Or simply the campaign stopped before the ponds formed? In my opinion, the differences in snowpack characteristics between the two years is very interesting, and very enlightening for modellers. Could the author include a discussion on slush layer and melt ponds (how they formed in 2016, and possibly why they did not form in 2015) in section 4.1?

---

## Author Comment (AC1) · 17 Oct 2019

Comment from author's Referee

My overall impression for the manuscript is that the first half part (âĹijSection 3.3) is detailed but the latter half part is redundant. Although Chapter 3 is "Results", causes of temporal variations of albedos are DISCUSSED with sensitivity tests using radiative transfer model in Section 3.4. In "4 Discussion", similar things as described in "3 Results" are described repeatedly.

Another reason I feel redundant is that each of Sections 3.4, 4.1, 4.2 and 4.3 and Chapter 5 consists of only one paragraph although those paragraphs have a volume more or less one page. These make the findings, discussing points and conclusion

of this paper ambiguous. It is needed to explain with a focus on important points and organize the text.

Author's response

The authors agree with the referee and find his comment pertinent. We voluntary decided to bring some discussion elements in section 3.4 (results) in order to introduce and justify, first the sensitivity study of SSA on albedo simulations (Figure 8), second the importance of the snow properties at the very surface (Figure 9) and third the effects of a varying ice albedo beneath the snowpack (Figure 10). We agree that these choices makes the discussion section redundant.

Changes in manuscript

The result section 3.4 will be reorganized. Only key elements needed to introduce the further snow albedo simulations (listed above) will remain in this section. The others elements will be moved to section 4.2. In addition, the discussion section will be divided into independent paragraphs in order to improve the readability and to point out the main focus and conclusions.

Comment from author's Referee

In Section 3.4, it is discussed on uncertainties in measurements of vertical snow physical properties as cause of disagreement in albedo between measurement and calculation in phases I and II. However, it is not discussed on the horizontal heterogeneity. This would be possible cause of the disagreement. Please discuss on this issue.

Author's response

We think that some main discrepancies between albedo measurements and simulations are indeed caused by high vertical heterogeneity of snow properties during phase I and II. During these periods spatial variability was very low. Especially during the surface melting period where surface changes (melt, fresh snow) were homogeneous over the snow cover. Remarkably, spatial variability was slightly higher during the first

phase. Generally, the thickest dunes showed lower albedo than places around because they were not covered by fresher windslabs due to their height. These considerations did not stand anymore for phase III where spatial variabilities played a larger role.

Changes in manuscript

The above details have to be added into the text.

Comment from author's Referee

p.5, L6-8 "The correction concerning the determination of SSA of wet snow introduced by Gallet et al. (2014b) was not applied in this study because it did not induce significant changes on albedo simulations at the end.": Please indicate amount of error in SSA by not applying this correction.

Author's response

The corrected SSA were calculated using high liquid water contents, higher than reasonably observed on the field. The corrections were not significant and did not explain the deviations between simulated and observed albedo.

Changes in manuscript

The amount of error in SSA by not applying this correction and the corresponding water contents used will be added to the text.

Comment from author's Referee

p.6 L4 "Albedo spectra were finally smoothed using a low-pass filter.": Does this mean the raw spectral albedo varies not smoothly unless a low-pass filter? Please explain the reason.

Author's response

The raw albedo spectra were indeed noisy. The noise was apparently related to the temperature of the spectrometer we used. Data from 2016 campaign were more affected because of the warmer conditions. In addition, we did not use the same spectrometer in both field campaign (but same brand and version).

Changes in manuscript

These details will be added to the text in section 2.5.

Comment from author's Referee

p.10, L22-25 "If it is the case, . . .": I don't understand this discussion on snow impurities. It is described "the impurities may have potentially lowered the albedo by 1% at maximum . . ." which is larger than the values of "reducing the reflected irradiance by 0.4% to 0.7%". If effects caused by snow thickness variations is larger than the impurities effect (1%), the latter (impurity effect) is not negligible.

Author's response

In the related section we aim to investigate the constant bias ($\sim$1.3 %) between the observed and simulated albedo, especially during phase I (where the measurements are the most trustful). We concluded that this bias is likely due to both, the shadows generated by the operator and the equipment and the presence of snow impurities. Since we focus our study on the effects of SSA and snow thickness, the latter largely affecting the albedo at shorter wavelength during phase III (Figure 10). It appears that the impurity content are negligible compared to snow thickness in this wavelength range and then are not incorporated into the modeling.

Changes in manuscript The text ill be arranged and clarified to avoid any misunderstanding from the reader.

Comment from author's Referee

4.1 Snowpack formation: This section describes observation results of snow physical properties qualitatively, some of which were shown in Sections 3.1 and 3.2. Dune formations continues from explanations for phases I – III in the first part, that is confusing.

At least discussion on dunes should be made in separate paragraph.

Author's response

The authors are very attached to the reconstruction of the snowpack history based on snow physical properties sampled months after snow layers formations. However, we agree that the corresponding paragraph is confusing and has to be reformulated.

Changes in manuscript

The section will be reorganized and divided into relevant paragraphs in order to improve the readability. In addition, redundant elements presents in both sections (result and discussion) will be reduced to a minimum.
* * *

---

## Author Comment (AC2) · 17 Oct 2019

Comment from author's Referee

Introduction should be compacted, leaving out all those details and concepts that deviate from the main scope of the paper.

From p.2 line 19 to p.3 line 17: what is the point here? Please move part of the text into the discussion section, to compare the results of this paper with previous ones, and leave in the introduction only the key message (few lines only), avoiding all the details and concepts that deviate from the focus of the paper. The introduction should explain the relevance of the treated issue and the problems that previous works have left unsolved and this paper will help to solve. Is the message here that the snow

processed studied in this work have been studied already a lot? What is then the added value of this work?

Author's response

We agree that too many details concerning snow metamorphism are given in the introduction.

Changes in manuscript

Details in the introduction will be reduced to a minimum. However, some information given are essential to propose a scenario of the snowpack formation for both years. Thus, snow metamorphism processes will be detailed when necessary in the discussion section.

Comment from author's Referee

The snow layer identification seems sometimes arbitrary. Due to the large spatial heterogeneity of the snowpack, it is hard to obtain the stratigraphic evolution of the snowpits, except in a broad line. The number of speculations on the evolution of the stratigraphy should be reduced to a minimum.

p.8, line 18-19: "Furthermore, layer II could be divided into two distinct layers of indurated faceted crystals which showed highest densities values, up to 500kgm−3, topped by a wind slab." Why the uppermost dune layer is called IIb instead of III? Isn't it generated by the same process at the same time?

Generally, it is very difficult to see correspondence between the layers described in fig 5 and the SSA and density profiles shown in Fig 6. The distinction between layer I and II is quite obscure in Fig 6 (do indurated depth hoar (layer I) and indurated faceted crystals (layer II) have same SSA and density?). Also the distinction between layer II and IV is hard to see in Fig 6. It seems to me that the schematic picture of the stratigraphy in fig 5 can be applies to few selected cases, but then most of the profiles are much more complex, also in view of the spatial heterogeneity. I therefore recommend

referring to Fig 5 only as an example of the observed stratigraphy, valid for a subgroup of snowpits, and mark in fig 6 what are the snowpits with stratigraphy that follows fig 5. Alternatively, I recommend to mark the 4 layers (e.g. with horizontal black lines) in each snowpit shown in fig 6. Actually, I recommend marking the layers also in case that stratigraphy is shown only for a selection of snowpits.

Author's response

The snow layer identification was based first on stratigraphic observations (snow grain shape) and then potentially on the SSA/density vertical profiles. The stratifications presented in Figure 5 summarized all the information sampled at the snowpits. Despite the spatial heterogeneity (snow depth and snow layer thickness) and the time evolution (snowfalls and melt), we believe that our summarized stratifications are relevant as we always observed the same snow layers in each snowpit or at least the bottommost layers. Concerning the five dunes, we decided to introduce layers IIa and IIb because, both are similar to layer II of conventional snowpits and present enough distinctions to justify a subdivision. This distinction was not observable for conventional snowpits maybe because this layer was much thinner than for dunes. It is true that the chosen stratigraphic layers presented in figure 5 do not match with the vertical profiles of SSA and density in Figure 6. It is because some stratigraphic layers, very different in term of grain shape, show very close values of SSA or density (for example layers I and II).

Changes in manuscript

We understand referee's comments, however we think that our stratifications should remain unchanged. The corresponding explanations in the text will be improved to avoid any misunderstanding from the reader. In addition, in Figure 6 dunes profiles will be labeled for a better recognition, and we will find a way to mark each layer (according to Figure 5) for every vertical profiles.

Comment from author's Referee

[Figure]

It is not always clear to which campaign (year) the results and discussion refer to. This should be clarified, and possibly the differences in snowpack characteristics between the two years should be discussed more extensively (personally I find the large observed differences very interesting, especially from modelling perspective).

Author's response

The large differences between the two years are indeed very interesting. Unfortunately, in 2016 much fewer snowpits were studied and the sampling began while snow had already experienced wet metamorphism at almost every depth.

Changes in manuscript

Years will be specified systematically and differences between 2015 and 2016 snowpacks will be briefly presented in section 4.1.

Comment from author's Referee

p. 7, line20: "several snowfalls". During the observing period in 2015, only two snowfall events were marked in Fig 4. In addition to these, where there other light snowfall events? If so, please mark all of them (see also my comments later)

Author's response

Changes in manuscript

Missing minor snowfalls will be specified in Figure 4.

Comment from author's Referee

Can the authors provide more descriptions of the melt ponds? How deep they were, where they open or frozen, how large they were, and were they covering the totality (or more than 90%) of field of view of the downward looking head of the spectro-radiometer (having, thus, a radius of more than 3m)? How many melt ponds have been measured? Had they varying characteristics? Are the provided values for bare ice and melt pond

albedo some averages? What was then the standard deviation? Also, in the discussion section it would be good to compare with the bare ice and pond albedo measurements carried out in previous studies.

One final question: is it so that melt ponds did not form in 2015? Or simply the campaign stopped before the ponds formed? In my opinion, the differences in snowpack characteristics between the two years is very interesting, and very enlightening for modellers. Could the author include a discussion on slush layer and melt ponds (how they formed in 2016, and possibly why they did not form in 2015) in section 4.1?

Author's response

Phase IV has been introduced to mark the snowpack vanishing and the profound changes in sea ice albedo. Properties of bar ice and melt ponds are not studied in detail because we focused on snow properties. In Figure 7, the provided albedo spectra are averages spectra of both surfaces (measurements were made over ice or ponds only). In 2015, ponds formed only few days before we left the ice camp. The pond formation was similar to 2016, ponds were first very extended before shrinking and settling.

Changes in manuscript

We think that we should not discuss of slush layers and ponds formation as many process behind remain unclear. We also lack of data that clearly link the snow cover, the ice properties and the slush layers or ponds locations.

Comment from author's Referee

p.10, lines 18-19: "Simulations with SSA reduced by 20% (see Figure 8 and Table 3), larger than the expected uncertainty, is not sufficient to offset the bias which is lowered to 1.0% at 500 nm." Based on the personal experience of some colleagues, who found strong overestimation of SSA using a similar measurement principle (IceCube), I think that in melting conditions the error in SSA measurements done with the applied sampling technique is closer to 100% than to 20%. The fact that the error increases with increasing wetness of the snow, and that it is most severe for the 1000nm than for the 500nm albedo, suggests that there is a problem in the SSA estimation. I recommend replacing the albedo modelled using +-20% SSA with the albedo modelled using + 100% SSA.

Author's response

Simulations with SSA reduced by 20% were made in order to investigate a potential bias in our SSA measurements. The results show that such a bias (far above the DUFISSS uncertainty) cannot explain the discrepancies between measurements and simulations, particular during phase I when snow was completely dry. It is true that melting conditions affected the SSA measurements during phase II and caused the largest discrepancies between measurements and simulations as discussed in section 4.3. However, those conditions did not prevail during this phase. For these reasons, the observed constant bias could not have been induced by erroneous SSA measurements.

Changes in manuscript

The argumentation showing that the constant discrepancies between measurements and simulations are not induced by a constant bias on SSA measurements should be based on observations during phase I. Observations during phase II were potentially affected by melting and then should be dismissed in this case.

Comment from author's Referee

p.11, line10-11: "Albedo spectra of bar ice and melt pond (as shown in Figure 7d) were used as soil albedo in TARTES." There cannot be snow above an open melt pond. Did the measured albedo spectra correspond to frozen melt ponds or to open melt ponds? They have quite different albedo. Did the authors measured the albedo of slush? Is it equal to the applied albedo of melt pond? These model results may be used to

interpret the evolution of albedo during phase III -IV in 2016, but not in 2015 (which is the year illustrated in Fig 6 and 8), where neither slush nor melt ponds were observed.

Author's response

The measured albedo correspond to open melt ponds. The aim of the study is to investigate the effects of the widest range of "soil albedo" beneath the snowpack on surface albedo. This is why we used the albedo of both bar ice and open melt pond. In 2015, a slush layer was observed in some cases, but as the snowpack was deep enough, it did not affect significantly the surface albedo ant then was not taken into account for simulation.

Changes in manuscript

The text will be clarified as described above.